# A rare gain of function mutation in a wheat tandem kinase confers resistance to powdery mildew

Ping Lu[1], Li Guo[2], Zhenzhong Wang[2], Beibei Li[1,3], Jing Li[4], Yahui Li[5], Dan Qiu[5], Wenqi Shi[6], Lijun Yang[6], Ning Wang[1,3], Guanghao Guo[1,3], Jingzhong Xie [1], Qiuhong Wu[1], Yongxing Chen[1], Miaomiao Li[1], Huaizhi Zhang[1,3], Lingli Dong[1], Panpan Zhang[1,3], Keyu Zhu[1,3], Dazhao Yu[6], Yan Zhang[7], Karin R. Deal[8], Naxin Huo[9], Cuimin Liu [1,3], Ming-Cheng Luo [8], Jan Dvorak [8], Yong Qiang Gu[9], Hongjie Li[5]* & Zhiyong Liu [1,3]*

Powdery mildew, caused by *Blumeria graminis* f. sp. *tritici* (*Bgt*), is one of the most destructive diseases that pose a great threat to wheat production. Wheat landraces represent a rich source of powdery mildew resistance. Here, we report the map-based cloning of powdery mildew resistance gene *Pm24* from Chinese wheat landrace Hulutou. It encodes a tandem kinase protein (TKP) with putative kinase-pseudokinase domains, designated WHEAT TANDEM KINASE 3 (WTK3). The resistance function of *Pm24* was validated by transgenic assay, independent mutants, and allelic association analyses. Haplotype analysis revealed that a rare 6-bp natural deletion of lysine-glycine codons, endemic to wheat landraces of Shaanxi Province, China, in the kinase I domain (Kin I) of WTK3 is critical for the resistance function. Transgenic assay of WTK3 chimeric variants revealed that only the specific two amino acid deletion, rather than any of the single or more amino acid deletions, in the Kin I of WTK3 is responsible for gaining the resistance function of WTK3 against the *Bgt* fungus.

[1] State Key Laboratory of Plant Cell and Chromosome Engineering, Institute of Genetics and Developmental Biology, The Innovative Academy of Seed Design, Chinese Academy of Sciences, Beijing 100101, China. [2] College of Agronomy and Biotechnology, China Agricultural University, Beijing 100193, China. [3] University of Chinese Academy of Sciences, Beijing 100049, China. [4] Plant Science and Technology College, Beijing University of Agriculture, Beijing 102206, China. [5] The National Engineering Laboratory of Crop Molecular Breeding, Institute of Crop Sciences, Chinese Academy of Agricultural Sciences, Beijing 100081, China. [6] Institute of Plant Protection and Soil Science, Hubei Academy of Agricultural Sciences, Wuhan 430064, China. [7] College of Horticulture, China Agricultural University, Beijing 100193, China. [8] Department of Plant Sciences, University of California at Davis, Davis, CA 95616, USA. [9] USDA-ARS West Regional Research Center, Albany, CA 94710, USA. *email: lihongjie@caas.cn; zyliu@genetics.ac.cn

Wheat (*Triticum aestivum* L.) provides 20% of the total daily calories and protein consumed by humankind[1]. Diseases and pests pose a serious threat to the global production of wheat grain and food supply[2]. Wheat powdery mildew is an epidemic disease caused by the biotrophic fungus *Blumeria graminis* f. sp. *tritici* (*Bgt*), which has resulted in severe yield losses during the past decades in China and other parts of the world[3,4]. Due to the rapid evolution of the pathogen and emergence of new virulent isolates, many race-specific resistance genes have become ineffective[4,5]. Development of wheat cultivars with new resistance genes to powdery mildew is thus an important breeding objective.

Although more than a hundred powdery mildew (*Pm*) resistance genes/alleles in 63 loci (*Pm1-Pm66*) have been documented[6,7], only seven of them, *Pm2*[8], *Pm3*[9], *Pm8*[10], *Pm21*[11,12], *Pm38/Yr18/Lr34/Sr57*[13], *Pm46/Yr46/Lr67/Sr55*[14], and *Pm60*[15] have been cloned and characterized so far. Most of these isolated *Pm* genes in wheat encode coiled coil nucleotide-binding leucine-rich-repeat (NLR) proteins. Resistance conferred by the NLR proteins tends to be overcome by fast evolution of virulent *Bgt* isolates, particularly when the gene is widely deployed in agriculture. The demise of resistance provided by the wheat-rye (*Secale cereale* L.) T1BL·1RS translocation carrying *Pm8*[3–5,16] exemplifies this problem. The partial resistance genes *Pm38/Yr18/Lr34/Sr57* and *Pm46/Yr46/Lr67/Sr55* encode an ATP-binding cassette (ABC) transporter[13] and a hexose transporter involved in sugar uptake[14], respectively. Those genes have provided durable resistance to powdery mildew as well as stripe rust, leaf rust, and stem rust (caused by *Puccinia striiformis* Westend. f. sp. *tritici* Eriks., *P. triticina* Eriks., and *P. graminis* Pers.:Pers. f. sp. *tritici* Eriks. & E. Henn., respectively) for decades.

Wheat landraces represent a rich source of disease resistance genes. For example, *Pm5d*[17], *Pm5e*[18], *Pm24*[19], *Pm24b*[20], *Pm45*[21], *Pm47*[22], and *Pm61*[23] were identified in Chinese landraces, and *Pm59*[24] and *Pm63*[25] were derived from landraces indigenous to Afghanistan and Iran, respectively. However, none of those genes have been cloned. This limits the understanding of their molecular basis and deployment in wheat breeding via molecular marker-assisted selection (MAS) and genome editing.

Powdery mildew resistance genes *Pm24*, *Pm24b*, and *MlHLT* were identified in Chinese wheat landraces Chiyacao (CYC)[19], Baihulu (BHL)[20], and Hulutou (HLT)[26], respectively. They were mapped to the same genomic region of the wheat chromosome arm 1DS.

In the present study, we report map-based cloning of *Pm24* and the characterization of its natural variation in conferring powdery mildew resistance. *Pm24* was found to be a rare natural allele of tandem kinase protein (TKP) with putative kinase-pseudokinase domains. A 6-bp deletion at the kinase domain of the wheat tandem kinase gene was found to be critical for the gain of function of powdery mildew disease resistance.

## Results

**Allelism at the *Pm24* locus.** To test the allelism of *Pm24*, *Pm24b*, and *MlHLT*, reciprocal crosses were made between landraces CYC, BHL, and HLT, and the $F_2$ populations were phenotyped using *Bgt* isolate E09 under controlled greenhouse conditions. All the $F_2$ plants from the crosses HLT × CYC (4,517 plants), HLT × BHL (4,318 plants), CYC × HLT (4,382 plants), CYC × BHL (4,590 plants), and BHL × HLT (4,692 plants) were resistant, indicating that *Pm24*, *Pm24b*, and *MlHLT* are tightly linked or allelic. Moreover, CYC, BHL, and HLT were highly or moderately resistant (Infection type 0-2) to 93 genetically divergent *Bgt* isolates collected from 45 counties in 12 provinces of China[27] (Supplementary Fig. 1, Supplementary Table 1). These results and

the nearly identical reactions of the three landraces to the pathogens demonstrate that *Pm24*, *Pm24b*, and *MlHLT* are most likely the same gene responsible for the resistance to powdery mildew.

**High-resolution mapping and map-based cloning of *MlHLT*.** The *MlHLT* locus was previously mapped to a 3.6 cM interval flanked by sequence tagged site (STS) markers *Xwggc3026* and *Xwggc3148* in the terminal region of chromosome arm 1DS[26] using a mapping population developed between the powdery mildew resistant HLT and highly susceptible cultivar Shi 4185 (S4185) (Fig. 1a, b). S4185 was highly susceptible to *Bgt* isolate E09 with a large number of visible conidia produced at 10 dpi (Infection type IT 4) and HLT was highly resistant to *Bgt* isolate E09 with no visible conidia produced (IT 0) (Fig. 1a). Trypan blue and DAB staining also showed large number of spores produced in S4185, but very mild cell death and robust accumulation of $H_2O_2$ in HLT (Fig. 1a). The two flanking STS markers were used in the BLAST homology search against the genome sequence of *Aegilops tauschii*[28], the diploid progenitor of the wheat D genome. The corresponding genomic sequences were subsequently used to develop closer markers linked to *MlHLT* (Supplementary Table 2). Two flanking simple sequence repeat (SSR) markers, *WGGB240* and *WGGB245*, were developed and used to screen a mapping population of 3,720 $F_2$ plants (7,440 gametes) derived from the cross of HLT × S4185 and 36 recombinants were identified. The closest SSR marker *WGGB241* and single nucleotide polymorphism (SNP) marker *WGGB244* placed *MlHLT* within a 0.06 cM genetic interval, corresponding to a 532 kb genomic region (Fig. 1c). Within this interval, the resistance phenotype co-segregated with STS marker *WGGB242* and derived cleaved amplified polymorphic sequence (dCAPS) marker WGGB243. Annotation of this 532 kb region identified two receptor-like kinase (RLK) genes *RLK1* and *RLK2*, a protein with two putative tandem kinase domains designated Wheat Tandem Kinase 3 (*WTK3*), a coiled-coil nucleotide-binding site with leucine-rich repeat (*CNL*) gene, a hypothetical protein (*HP*), and a 50 S ribosomal protein L10 (*RP*) (Fig. 1e, f).

The expression profiling indicated that genes *WTK3*, *RLK2*, *HP*, *CNL*, and *RP* were expressed in the seedlings of both HLT and S4185 (Supplementary Fig. 2a). To identify the candidate resistance genes, the genomic DNAs of these expressed genes were cloned from HLT and S4185 for sequence comparison. SNPs or insertion-deletions (InDels) were identified only for *WTK3* and *CNL* between HLT and S4185 (Fig. 1g, h).

The full-length cDNAs of *WTK3* and *CNL* were obtained from HLT and S4185 by the 5′- and 3′-rapid amplification of complementary DNA (cDNA) ends (RACE). In the resistant landrace HLT, the *WTK3* gene was 10,410 bp from the start to the stop codons and contained 11 exons with a coding sequence of 2682 bp. It encodes a protein containing 893 amino acids with two putative tandem protein kinase domains. A C/G SNP in the third intron and a 6-bp deletion (5880-AAAGGA-5881) in the fifth exon of *WTK3* were detected between the sequences from HLT and S4185, leading to a lysine-glycine two-amino acid (K400G401) deletion in the encoded WTK3 protein of HLT (Fig. 1g). The *CNL* gene in HLT is 4164 bp long with 3 exons and the coding sequence is 3117 bp. It encodes a RPM1-like protein of 1,038 amino acids with three highly conserved domains, including one RX-cc-like domain and two NB-ARC domains. There were a nonsynonymous SNP (T1613C) in the second exon and a single nucleotide deletion (G3470-) in the 3rd exon of the *CNL* gene, resulting in missense and nonsense mutations in the CNL protein of S4185, respectively (Fig. 1h). Since *WTK3* and *CNL* are known to play important roles in plant immunity, they

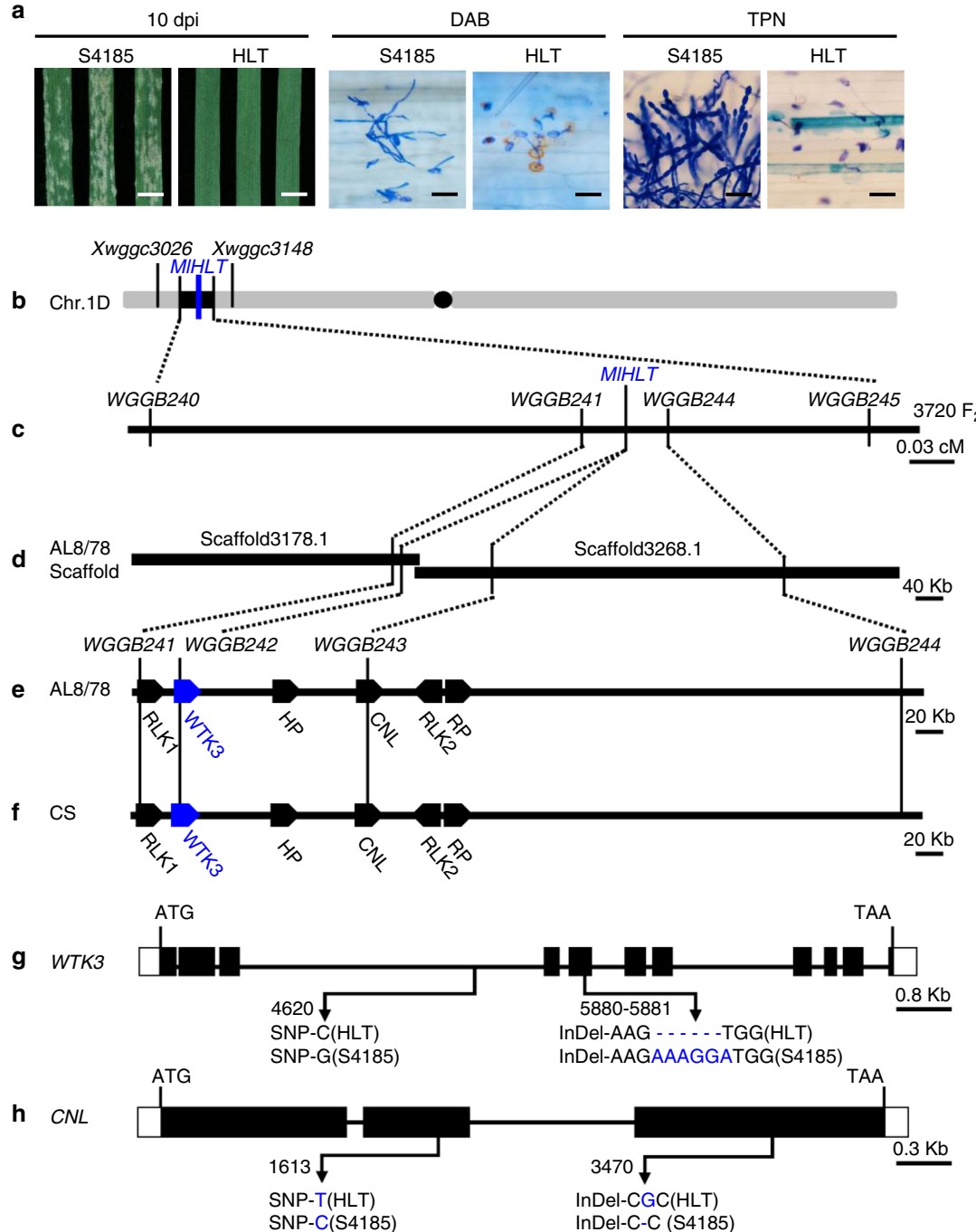

**Fig. 1 Map-based cloning of *MlHLT*. a** Chinese wheat landrace HLT is highly resistant to powdery mildew. Two-week-old S4185 and HLT plants were inoculated with *Bgt* isolate E09. Representative leaves were photographed at 10 d post inoculation (dpi). Bar, 5 mm. DAB staining of leaves infected with *Bgt* isolate E09 at 2 dpi. Brown staining shows the accumulation of $H_2O_2$. Bar, 100 μm. Trypan blue staining of the leaves infected with *Bgt* isolate E09 at 7 dpi to visualize fungal structures and plant cell death. Scale bar, 100 μm. **b** Genomic region containing *MlHLT* on the short arm of wheat chromosome 1D. **c** Genetic linkage map of *MlHLT* generated using a mapping population of 3,720 $F_2$ plants derived from the cross of HLT × S4185. **d** AL8/78 scaffolds of the *MlHLT* region anchored to the genetic linkage map. **e** Predicted genes of AL8/78 in the *WGGB241-WGGB244* interval. **f** Predicted genes of Chinese Spring (IWGSC RefSeq v1.0) in the *WGGB241-WGGB244* interval. Receptor-like kinase (*RLK1* and *RLK2*), wheat tandem kinase 3 (*WTK3*), coiled-coil nucleotide-binding site with leucine-rich repeat (*CNL*), hypothetical protein (*HP*), and 50 s ribosomal protein L10 (*RP*) genes were annotated in this interval. **g**, **h** Difference in the structures and genomic sequences of *WTK3* and *CNL* genes between HLT and S4185, respectively.

were both considered as the candidate genes of *MlHLT* for further functional characterizations.

**Validation of *WTK3* and *CNL* by transgenic assay.** Two constructs, *ProWTK3:WTK3* and *ProCNL:CNL*, driven by the native promoters of *WTK3* and *CNL* genes from HLT, were separately delivered into the susceptible hexaploid wheat cultivar Fielder by *Agrobacterium*-mediated transformation. The *ProWTK3:WTK3* construct consisted of a 15,599 bp genomic fragment that included a 3378 bp presumed native promoter, the 10,410 bp entire gene body including coding and intron region, and a 1811 bp

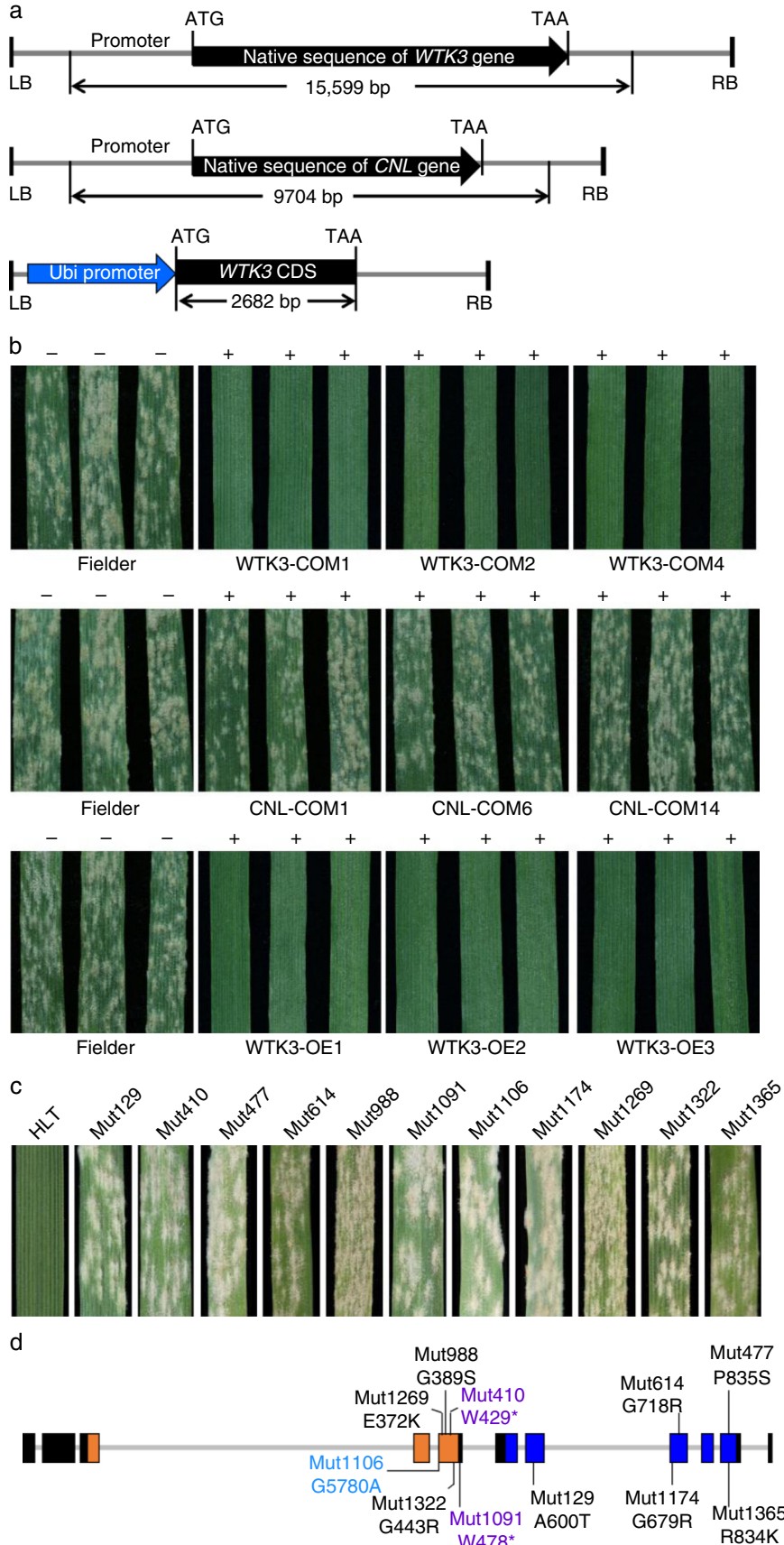

**Fig. 2 Validation of *MlHLT* candidates using transgenic assay and EMS mutants. a** Structure of *ProWTK3:WTK3*, *ProCNL:CNL*, and *ProUbi:WTK3* used for transgenic assays, respectively. The *ProWTK3:WTK3* construct contains the entire *WTK3* gene, 3,378 bp promoter, and 1,811 bp terminator. The *ProCNL:CNL* construct contains the entire *CNL* gene, 3,064 bp promoter, and 2,476 bp terminator. *Ubi*, promoter of the maize polyubiquitin gene. LB, left border; RB, right border. **b** Infection reactions of Fielder (Infection type IT 4), the $T_1$ transgenic plants of *ProWTK3:WTK3* (IT 0), *ProCNL:CNL* (IT 4), and *ProUbi:WTK3* (IT 0) to *Bgt* isolate E09, respectively. Three representative individuals of each transgenic line were photographed at 10 dpi; $+/-$: presence/absence of the transgene. **c** Powdery mildew resistance assessments of *WTK3* EMS mutants. Two-week-old HLT (IT 0) and 11 mutants (IT 3-4) were inoculated with *Bgt* isolate E09. Representative leaves were photographed at 10 dpi. **d** EMS mutants carrying single nonsense or missense mutations in the *WTK3* gene sequences. Structure of gene *WTK3* (from the start to the stop codon) was presented. Gray straight lines indicate introns, and rectangles indicate coding exons (orange and dark blue rectangles represent the Kin I and Kin II domains, respectively). The positions of the mutations are indicated by black lines. Mutation names in purple, black, and blue indicate nonsense, missense, and splice site mutations, respectively.

terminator region of the *WTK3* gene (Fig. 2a). The *ProCNL:CNL* construct contained a 9,704 bp genomic fragment with 3064 bp presumed native promoter, 4164 bp exon and intron regions, and 2476 bp terminator (Fig. 2a). The transformation of the *WTK3* construct yielded 7 positive $T_0$ plants with the confirmed transgene sequence. The positive $T_0$ individuals were advanced to produce the $T_1$ generation. All plants carrying the *ProWTK3:WTK3* construct in the 7 independent $T_1$ transgenic families were highly resistant to *Bgt* isolate E09 infection (Fig. 2b, Supplementary Table 3). In contrast, all plants positive for *ProCNL:CNL* construct in the 14 independent $T_1$ transgenic families were highly susceptible to the same *Bgt* isolate (Fig. 2b, Supplementary Table 3).

The full-length *WTK3* cDNAs from HLT had three alternative splicing variants: WTK3$^{HLT}$.IF1, WTK3$^{HLT}$.IF2, and WTK3$^{HLT}$.IF3. WTK3$^{HLT}$.IF1 was found to be the main isoform (41 out of 46 tested WTK3 cDNA clones), while the WTK3$^{HLT}$.IF2 and WTK3$^{HLT}$.IF3 were much less abundant (6.52% and 4.34%, respectively). WTK3$^{HLT}$.IF1 encodes a full-length intact WTK3 protein with the putative kinase I (Kin I) and kinase II (Kin II) domains, whereas WTK3$^{HLT}$.IF2 and WTK3$^{HLT}$.IF3 encodes proteins with the truncated Kin I domain only (Supplementary Data 1).

The alternative transcripts of susceptible allele *WTK3* in cultivar S4185 also had three alternative splicing isoforms: WTK3$^{S4185}$.IF1, WTK3$^{S4185}$.IF2, and WTK3$^{S4185}$.IF3. The WTK3$^{S4185}$.IF1 was the main transcript (95% of the tested *WTK3* cDNAs) encoding a wheat tandem kinase protein including the putative Kin I and Kin II domains (Supplementary Data 1). An overexpression construct *ProUbi:WTK3* was generated to express WTK3$^{HLT}$.IF1 under the maize (*Zea mays* L.) Ubi promoter (Fig. 2a) in Fielder by *Agrobacterium*-mediated transformation. All plants of the 12 $T_1$ transgenic families positive for the construct were highly resistant to the *Bgt* isolate E09 (Fig. 2b, Supplementary Table 3). Taken together, these results indicate that the *WTK3* gene is sufficient to provide powdery mildew resistance as the causal gene in the *MlHLT* locus.

**Analysis of ethyl methanesulfonate mutant lines**. To further test if *WTK3* was required for resistance against the *Bgt* pathogen, we screened 1,360 $M_2$ lines of EMS-mutagenized HLT population with *Bgt* isolate E09 and identified 26 mutant families showing segregation for powdery mildew susceptibility (Fig. 2c). Sequencing of the *WTK3* and *CNL* genes (including the putative promoters, exons and introns, and terminator regions) in the susceptible plants of these $M_2$ families revealed the presence of mutations only in the *WTK3* but not in the *CNL*. Eleven of these independent susceptible mutants had SNPs that resulted in amino acid substitutions, premature stop codons, or relocation of the intron/exon splice sites (Fig. 2d, Supplementary Table 4). Two mutants, Mut410 and Mut1091, were nonsense mutations that gave rise to premature stop codons at the encoded amino acid positions of 429 and 478, respectively. A frameshift mutation was

detected in Mut1106 with a G/A point mutation in the splice acceptor site of intron 4. The other eight mutants harbored missense mutations that occurred in several putative conserved subdomains of the Kin I and Kin II domains, indicating that the two putative kinase domains of *WTK3* are essential for *MlHLT* to function the resistance against the powdery mildew disease. Loss of function mutations that occur in the kinase domains of *WTK3* gene provides additional evidence that the *WTK3* gene is *MlHLT*. No sequence variation was detected in the *WTK3* and *CNL* genes (including the putative promoters, exons and introns, and terminator regions) of the other 15 EMS mutants, suggesting possible mutations in other unknown genes or elements involved in the *MlHLT* regulation pathway.

**Expression pattern of *WTK3***. RT-PCR analyses demonstrated that *WTK3* was expressed in wheat root, stem, and leaf tissues (Supplementary Fig. 2b). In the seedling leaves of both HLT and S4185, expression of *WTK3* showed a 6-fold up-regulation and peaked at 24 h post inoculation (hpi) with *Bgt* isolate E09, when compared to the un-inoculated plants (Supplementary Fig. 2c). However, higher expression of WTK3$^{HLT}$ than that of WTK3$^{S4185}$ ($P < 0.01$, Students *t*-test, $n = 3$) was only observed at 36 hpi. In addition, the expression patterns of six pathogenesis-related (PR) genes (i.e., *PR1*, *PR2*, *PR3*, *PR4*, *PR5*, and *PR9*) involved in plant defenses were investigated at the RNA expression level as well. Significant up-regulations of the *PR* genes were observed in the leaves of the resistant HLT than the susceptible S4185 upon inoculation with the *Bgt* isolate E09 (Supplementary Fig. 2d). Compared to the *WTK3*, the up-regulations of the *PR* genes were at later time points, implying that *WTK3* should be an upstream regulator of the *PR* genes in the powdery mildew defense response pathway.

**Allelic variations of the *Pm24* locus**. To further test whether the powdery mildew resistance genes *Pm24*, *Pm24b*, and *MlHLT* are identical, gene-specific primers were designed to amplify the entire 15,599 bp genomic sequence, including the presumed promoter, exons and introns, and terminator region, of the *WTK3* gene from CYC (*Pm24*) and BHL (*Pm24b*). Sequence comparison revealed that CYC, BHL, and HLT shared an identical sequence in the *WTK3* genomic region and CDS (Supplementary Data 2), indicating that *Pm24*, *Pm24b*, and *MlHLT* are the same allele. To characterize the sequence variation of the entire *WTK3* gene body in wheat germplasm, *WTK3* was amplified and sequenced from 7 *Ae. tauschii* accessions (AL8/78, PI 431062, PI 486274, RM 000182, PI 511381, PI 486271, and PI 511367), 12 Chinese common wheat landraces (Mazhamai, Hongtoumai, Hongyouzi, Huixianhong, Baimangmai, Hongmai, Wangshuibai, Xiaofoushou, Chinese Spring, Hanzhongbai, Zijiehong, and Baihuamai), and one modern cultivar Aikang 58. Sequence alignment identified many SNPs and InDels in the *WTK3* gene body among these accessions (Supplementary

Data 3). Based on the major sequence variations, the 24 accessions were classified into four major groups (Supplementary Fig. 3a). Group 1 contains 9 common wheat accessions including HLT, BHL, CYC, S4185, the other 4 landraces, and Aikang 58. Groups 2 and 3 include the *Ae. tauschii* accessions, and group 4 consists of the other 8 common wheat landraces including Chinese Spring. Compared to the other 3 groups, Group 1 has a 1,509 bp insertion in the seventh intron of *WTK3*. However, the 6-bp deletion at the fifth exon of *WTK3* was present only in CYC, BHL, and HLT of Group 1, but not in the other accessions of Group 1 to Group 4 (Supplementary Data 3, Supplementary Data 4).

To further understand the sequence variations of the 6-bp InDel and confirm its association with powdery mildew resistance, STS marker *InDel-WTK3* amplified a 632 bp fragment covering the 6-bp InDel region in HLT was used to genotype 1,069 accessions of geographically diverse worldwide common wheat collections and *Ae. tauschii* accessions. Sequence alignments revealed two InDels, the 6-bp InDel (5880-AAAGGA-5881) at the fifth exon and the 114 bp InDel (5906-ATT…TAA-5907) at the fifth intron of *WTK3*, in this genomic region. In addition, multiple SNPs were also detected in the amplified region among these accessions. Based on the sequence variations of this genomic region, *WTK3* could be divided into 10 haplotypes (i.e., Hap I to Hap X) (Supplementary Fig. 3b, Supplementary Fig. 4). Hap I includes CYC, BHL, HLT, and another Chinese wheat landrace, Hongmangmai (HMM, ZM003894), all sharing an identical *WTK3* sequence with the 6-bp deletion (Supplementary Data 2, Supplementary Data 3) and highly resistant to 36 tested *Bgt* isolates (Supplementary Fig. 1, Supplementary Table 1). Moreover, the *WTK3* full genomic DNA and CDS of HMM were identical with the *WTK3* sequences of CYC, BHL, and HLT (Supplementary Data 2). The 6-bp deletion in *WTK3* was not detected in any of the remaining 1,065 common wheat and *Ae. tauschii* accessions (including 891 powdery mildew-susceptible accessions), indicating that this 6-bp deletion is a rare gain-of-function natural variation unique for *Pm24* in the Chinese wheat landraces that is associated with powdery mildew resistance (Supplementary Data 4). Based on the 6-bp deletion, a *Pm24* gene-specific STS marker *STS-Pm24* was designed for effective detection of the target gene in molecular marker-assisted selection (MAS) of *Pm24* (Supplementary Fig. 5).

**The 6-bp deletion in *WTK3* is critical for resistance**. The difference in the 6-bp deletion between the resistant landrace HLT and the susceptible cultivar S4185 is located in the Kin I domain of *WTK3*. The two-amino acid lysine-glycine (K400G401) deletion from the WTK3$^{HLT}$ protein was not located in the key conserved key residues of the Kin I domain (Supplementary Fig. 6). To further understand the association of the 6-bp deletion with the powdery mildew resistance function of *WTK3*, three chimeric variants of *WTK3* (i.e., *ProUbi*:WTK3#1, *ProUbi*: WTK3#2, and *ProUbi*:WTK3#3) were generated with a different 3-bp (AAA), 3-bp (GGA), and 12-bp (AAGAAAGGATGG) deletion, respectively, in the corresponding 6-bp (AAAGGA) deletion region, and overexpressed them individually in the susceptible cultivar Fielder (Fig. 3a, Supplementary Fig. 7). Compared to the susceptible allele WTK3$^{S4185}$, WTK3#1 and WTK3#2 were 894 amino acids in size with only one amino acid deletion at K400 (lysine) and G401 (glycine), respectively, and WTK3#3 consisted of 891 amino acids with a deletion of 4 amino acids (K399_W402, lysine-lysine-glycine-tryptophan) (Supplementary Fig. 7). All of the positive $T_1$ transgenic plants overexpressing *WTK3#1*, *WTK3#2*, and *WTK3#3* were highly susceptible to *Bgt* isolate E09 (Fig. 3b, Supplementary Table 3),

demonstrating that the specific deletion of the two amino acid lysine-glycine (K400 and G401) at the Kin I domain of WTK3 is critical for gaining the powdery mildew resistance function.

**Discussion**

In plants, NLR proteins and protein kinases (PKs) are the major classes of disease resistance genes. NLR functions as intracellular immune receptor that recognizes pathogen effectors and activates effector-triggered immunity (ETI)[29] and protein kinases are important for transmembrane signaling that regulates plant development and adaptation to diverse environmental conditions[30]. Except for the pleiotropic partial resistance genes *Pm38/Yr18/Lr34/Sr57*[13] and *Pm46/Yr46/Lr67/Sr55*[14], most cloned powdery mildew resistance genes in wheat so far encode NLR proteins that tend to be race-specific[8–12,15]. *Pm24* (*WTK3*) from Chinese wheat landrace Hulutou encodes a serine/threonine non-arginine-aspartate (non-RD) receptor-like protein kinase with two putative tandem kinase domains (Kin I and Kin II) and without extracellular domain.

Receptor-like cytoplasmic kinases (RLCKs) usually function at the plasma membrane during plant pattern recognition receptor (PRR)-mediated immunity. Tomato (*Solanum lycopersicum* L.) *Pto* gene is the first characterized RLCK encoding a serine/threonine protein kinase that confers resistance to races of *Pseudomonas syringae* pv. *tomato*[31]. The *Arabidopsis Botrytis*-induced kinase 1 (BIK1) is another well-characterized RLCK that plays an important role downstream of the PRR-brassinosteroid-insensitive 1 (BRI1)-associated kinase 1 (BAK1) complex by direct interaction with BAK1 in the resistance to fungal pathogen *Botrytis cinerea*[32]. The RLCK protein BR-SIGNALING KINASE 1 (BSK1) was found to be involved in *Arabidopsis* powdery mildew resistance by positively regulating pathogen-associated molecular pattern (PAMP)-triggered immunity (PTI), which is the first layer of plant defense against invading pathogens[33]. A serine/threonine protein kinase Stpk-V isolated from the short arm of chromosome 6 V of *Haynaldia villosa* in the T6VS·6AL translocation is a key member of the *Pm21* gene regulation pathway that confers durable and broad-spectrum resistance to wheat powdery mildew[34]. The isolation of WTK3 with the tandem kinase domains provides opportunity for understanding the signal transduction function of tandem kinase protein (TKP) in wheat powdery mildew resistance and plant innate immunity.

Recently, TKP has emerged as a new class of disease resistance kinase protein family in plant innate immunity. Four plant disease resistance genes have been identified to contain a structure with tandem kinase domains, i.e., wheat stripe rust resistance gene *Yr15* (*WTK1*)[35], wheat stem rust resistance gene *Sr60* (*WTK2*)[36], barley (*Hordeum vulgare* L.) stem rust (*P. graminis* f. sp. *tritici*, *Pgt*) resistance gene *Rpg1*[37], and the candidate gene (*MLOC_38442.1*) of barley loose smut (*Ustilago nuda*) resistance gene *Un8*[38]. WTK3 is a new member of the TKP family conferring resistance to powdery mildew in wheat. Based on the sequence conservation of the key amino acid residues for kinase function in the two kinase domains, WTK1 can be classified as tandem kinase-pseudokinase, WTK2 and Un8 are tandem kinase-kinase, while RPG1 is a tandem pseudokinase-kinase. Kinase domain sequence comparison revealed that the amino acid residues in the core of kinase catalytic domain of WTK3 were conserved in the Kin I domain as that of BIK1, Pto, Stpk-V, RPG1, WTK1, and WTK2 (Supplementary Fig. 6), suggesting that the Kin I domain of WTK3 should be a putative functional kinase. Two EMS-induced susceptible mutants, Mut1269 and Mut988, resulted in nonsynonymous missense mutations on the key residues of subdomains VIII (E372K) and IX (G389S) of Kin I, implying the function of Kin I of WTK3 is critical to powdery

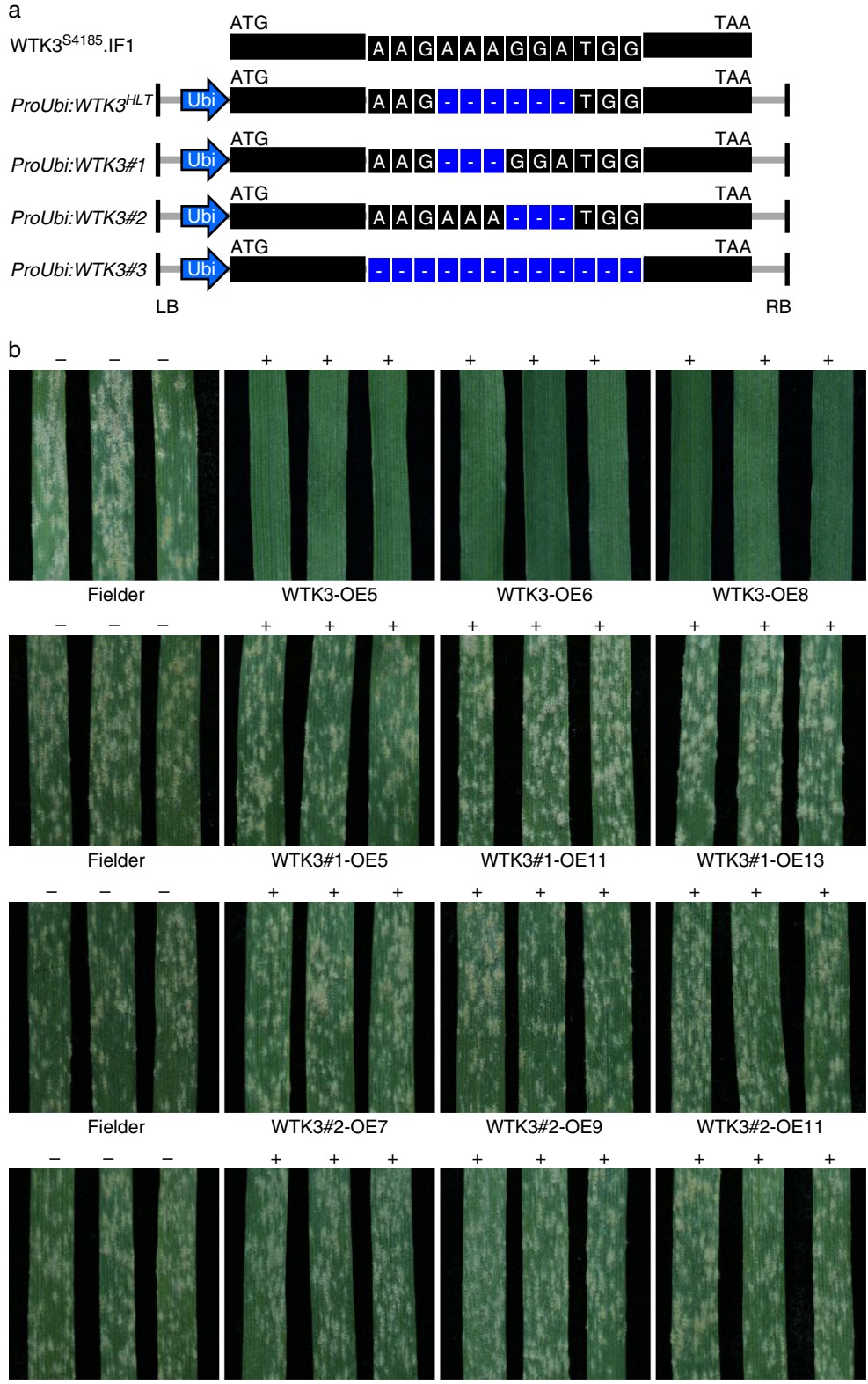

**Fig. 3 The 6-bp deletion in the fifth exon of *WTK3* is critical for the powdery mildew resistance of *Pm24*. a** The structures of the main transcript of *WTK3* in S4185, and the constructs of *WTK3* overexpression variants *ProUbi:WTK3*HLT, *ProUbi:WTK3#1*, *ProUbi:WTK3#2*, and *ProUbi:WTK3#3* used for transgenic assay. *Ubi*, promoter of the maize polyubiquitin gene. LB, left border; RB, right border. **b** Infection reactions of Fielder (IT 4) and the T₁ transgenic plants of *ProUbi:WTK3*HLT (IT 0), *ProUbi:WTK3#1* (IT 4), *ProUbi:WTK3#2* (IT 4), and *ProUbi:WTK3#3* (IT 4) inoculated with *Bgt* isolate E09. Three representative individuals of Fielder and each transgenic line were photographed at 10 dpi; +/−: presence/absence of the transgene.

mildew resistance. The same key residues are more divergent in the subdomains I, VII, VIII, and especially the catalytic loop VIb of Kin II, indicating that the Kin II domain of WTK3 may be a pseudokinase as that of the Kin II of WTK1[35] and Kin I of Rpg1[37] (Supplementary Fig. 6). However, several EMS-induced mutations on or nearby the key residues of subdomains III (A600T), VII (G718R, left border of the activation loop), and XI (R834K and P835S) at the Kin II resulted in susceptibility to powdery mildew, suggesting that the Kin II of WTK3 is also important in resistance to the Bgt pathogen. Similar phenomenon was also observed in the barley stem rust resistance gene Rpg1. Even through the Kin I of Rpg1 was predicted as a pseudokinase, transgenic assays expressing either mutated Kin I or Kin II domains of Rpg1 resulted in fully susceptibility to Pgt, demonstrating the requirement of both kinase domains of Rpg1 for the disease resistance[39].

Yr15, Rpg1, and Un8 have been shown to provide a broad-spectrum of disease resistance to their respective biotrophic fungal pathogens[35,37,38]. However, the race-specific stem rust resistance gene Sr60 conferred intermediate level of resistance to 3 of 8 tested Pgt races by delaying but not stopping the Pgt infection; therefore it is classified as a partial resistance gene[36]. In the current study, we showed that the Pm24 carrying accessions CYC, BHL, HLT, and HMM are resistant to all of the tested 93 Bgt isolates collected from 12 provinces of China. With the present assay, we cannot formally rule out that there are Bgt isolates which actually overcome Pm24/WTK3, but which are held back by other resistance genes in the lines carrying Pm24/WTK3.

Phylogenetic analysis of 192 individual kinase/pseudokinase domains indicated that the Kin I domain of WTK3 is quite distant from the Kin I of WTK1 (Yr15) and Un8, but very close to the Kin I domain of WTK2 (Sr60) and Rpg1. However, the Kin II domain of WTK3 is close to those of Rpg1 and Un8, but distant from those of WTK1 and WTK2, indicating a different origin of these TKPs. These results suggested that WTK1, WTK2, and Un8 were derived from two kinase domain fusions while RPG1 and WTK3 originated from kinase domain duplications (Supplementary Fig. 8). The relatively high amino acid sequence identity (52.98%) between the Kin I and Kin II of Rpg1 suggested a relatively recent duplication of the two-kinase domains. However, the residues of one kinase domain has been changed tremendously after duplication and the Kin I and Kin II domains of WTK3 shared only 29.3% amino acid sequence identity, indicating that an relatively ancient duplication event occurred.

The susceptibility alleles of WTK3 were identified as TraesCS1D02G058900.1 in T. aestivum cv. Chinese Spring (AABBDD)[40] and AET1Gv20142700.38 in Ae. tauschii (DD)[28]. Homoeologs of WTK3 were found on the homoeologous chromosome 1BS of Chinese Spring (TraesCS1B02G075800.1) and durum wheat (T. durum, AABB) accession Svevo (TRITD1Bv1G020600.1)[41]. However, they were not detected in the chromosomes 1A and 1B of wild emmer wheat (T. turgidum ssp. dicoccoides) Zavitan genome[42] and the chromosome 1A of Chinese Spring, Svevo and T. urartu G1812 (AA)[43], suggesting that the orthologous copy of WTK3 may have been lost from those sub-genomes during wheat evolution. The WTK3 orthologs were also identified in the orthologous chromosomes of Hordeum vulgare (HH, HORVU1Hr1G011660.17)[44] and S. cereale (RR, Sc1Loc00250465.1)[45], indicating a high collinearity of WTK3 in the orthologous chromosomes of Triticeae species. A homologous copy of WTK3 was found in Brachypodium distachyon, but at a non-synteny chromosome region (Bradi5g01574.3)[46]. The homologous proteins of WTK3 identified in rice (Oryza sativa, XP_015697519.1), sorghum (Sorghum bicolor, XP_021317235.1), and maize (Z. mays, XP_008679435.2) were resolved as an outgroup (Supplementary Fig. 8). The presence of WTK3 homologs

in the grass family suggested an ancestral duplication of the WTK3 before the divergence of the Poacease species. In addition to CYC, BHL, and HLT, only one Chinese wheat landrace Hongmangmai (HMM) carries the Pm24/WTK3 gene after screening of 1,069 common wheat and Ae. tauschii accessions collected from geographically diverse regions worldwide. The allelism test and the identification of exact sequence of WTK3 in the four resistant Chinese wheat landraces revealed that Pm24, Pm24b, and MlHLT are the same gene. Geographically, BHL, HLT, and Hongmangmai were collected in Shaanxi province, China, while CYC was in a nearby Henan province. The very low percentage of WTK3 with a powdery mildew resistance functional allele (0.37%) in the worldwide wheat germplasm indicates that Pm24 is a rare gain of function natural allele that is most likely present only in the Chinese wheat landraces from Shaanxi province and its nearby region. This observation suggests that the Pm24 allele may originate recently as a natural mutation after common wheat was introduced into central China that has not been widely spread into other landraces and not been used in modern wheat breeding program.

A majority of plant RLCKs studied to date interacted with receptor kinases (RKs) and phosphorylation of RLCKs is essential for RLCK-mediated signaling pathway. Site-directed mutagenesis analysis of the kinase domains of Rpg1 indicated that the pseudokinase Kin I domain did not have auto-phosphorylation whereas the Kin II domain was active in auto-phosphorylation of serine and threonine residues in vitro[39]. Transgenic mutants encoding an RPG1 protein with an in vitro inactive kinase domain fail to phosphorylate RPG1 in vivo and are susceptible to stem rust, demonstrating that phosphorylation is a prerequisite for disease resistance. Protein kinase inhibitors result in susceptibility to stem rust by preventing RPG1 phosphorylation, providing further evidence for the importance of phosphorylation of TKP in disease resistance[47]. The specificity of the WTK3 allele to confer powdery mildew resistance hinges on deletion of two amino acids (K400G401) from the Kin I domain of WTK3. The lysine-glycine residues were not conserved residues of active kinase domain and were located beyond the core catalytic/activation loop (Supplementary Fig. 6); hence the potential kinase activity of WTK3 should not be affected.

Predicted structure of WTK3 based on the crystal structure of human pseudokinase/kinase protein from Jak-family member TYK2[48] and Arabidopsis BIK1[49], as well as the predicted 2D structure of WTK1[35] revealed that the K400G401 residues lie in the loop region between the two α-helixes, αF (subdomain IX), and αG (subdomain X) (Supplementary Fig. 9). Deletion of the two amino acids resulted in a more compact loop shorter than the other homologous kinases (Supplementary Fig. 6). Compared to the homologous kinase domains of BRI1 and BAK1, a uniquely extended "open flower" shaped loop in Arabidopsis BIK1 might provide an interface for protein-protein interaction involved in downstream of PRR signaling pathway[49]. Transgenic assay using the artificial chimeric variants with less or more residues deletion revealed only the lysine-glycine (K400G401) deletion, rather than any other deletions, is responsible for gaining the resistance function of WTK3 against the Bgt fungus. The critical two-amino-acid deletion in gaining of function of powdery mildew resistance implies that a precise protein-protein interaction may be involved in the WTK3 signaling pathway.

The TKP members PRG1, Un8, WTK1, WTK2, and WTK3 have been found to play specific roles during immunity against various fungal pathogens in barley and wheat. It will be interesting in the future to determine if other TKP members also involved in plant disease resistance, especially in the grass family. Characterization of the TKP kinase activities, associated proteins and phosphorylation processes would shed a light on

the understanding of molecular mechanism of this protein family in plant innate immunity. The isolation of *Pm24* further demonstrates the importance of crop landrace conservation. The absence of *Pm24* in modern wheat cultivars and the identification of the key 6-bp deletion associated with powdery mildew resistance provide a potential opportunity for developing disease resistant cultivars with MAS and precise genome-editing technologies in wheat breeding program.

## Methods

**Plant materials.** The $F_2$ mapping population was developed from a cross between the Chinese wheat landrace HLT and S4185, a Chinese elite winter wheat cultivar highly susceptible to powdery mildew. Plants were grown in a greenhouse with 16 h light/8 h dark (24/18 °C, 70% relative humidity). The powdery mildew susceptible soft white spring wheat cultivar Fielder was used for the transgenic experiment. Wheat collections for haplotype analysis included 262 entries from Chinese wheat mini-core collection (MCC)[50], 371 and 347 common wheat landraces from China and the other 34 countries, respectively, two modern cultivars S4185 and Aikang 58, and 87 accessions of *Ae. tauschii* (Supplementary Table 5). Common wheat cultivars Xuezao and Chancellor were used as the susceptible controls in the powdery mildew assessments.

**Pathogen inoculation.** *Bgt* isolate E09[25] and another 92 genetically divergent isolates from different 12 provinces of China[27] were used for powdery mildew evaluations (Supplementary Fig. 1, Supplementary Table 1). Wheat seedlings at the two-leaf stage (two-week-old) were inoculated with *Bgt* isolates as previously described[26,27]. Three individual plants of CYC, BHL, HLT, HMM, and Chancellor were inoculated with 36 *Bgt* isolates (Supplementary Fig. 1). At least 20 individuals of the $T_1$ transgenic families were inoculated with *Bgt* isolate E09. For fine mapping of *MlHLT*, 25-30 plants of the $F_3$ families were screened for reaction to *Bgt* isolate E09 to confirm the genotypes of the corresponding $F_2$ plants. Disease symptoms were recorded 10 d after inoculation using a scale from 0 to 4 infection type (IT) (0; for necrotic flecks, and 1 − 4 for highly resistant, moderately resistant, moderately susceptible and highly susceptible[26]. *Bgt* isolates were maintained and increased on the highly susceptible cultivar Xuezao or Chancellor seedlings.

**Detection of $H_2O_2$ reaction and plant cell death.** To detect the accumulation of $H_2O_2$, the first leaves were cut from plants of HLT and S4185 at 48 h after inoculation with *Bgt* isolate E09 and were incubated in a 3,3′-diaminobenzidine (DAB) solution (1 mg/mL, pH 5.8) for 12 h, and then bleached in absolute ethanol[51]. Before assessing the accumulation of $H_2O_2$, the bleached leaves were incubated in a 0.6% (w/v) Coomassie blue solution for 10 s and then washed with water. To detect plant cell death, 7 d after *Bgt* isolate E09 inoculation, the primary leaves from HLT and S4185 plants were incubated in a 0.4% Trypan blue solution for 5 min in boiling water, bleached for 24 h in chloral hydrate solution (5:2, w/v), and stained in a 0.6% (w/v) Coomassie blue solution for 10 s[52]. The treated leaves were observed under an Olympus BX-53 microscope.

**Mutagenesis and mutation screening.** Seeds of HLT (15,000) were soaked in $H_2O$ for 8 h, treated with a 0.5% (v/v) EMS solution by incubating on a shaker at 100 rpm at room temperature for 16 h, and washed with running water at room temperatures for 4 h[53,54]. The mutagenized seeds were grown in the field and 1,360 plants were obtained in the $M_1$ generation. Seeds were harvested at maturity and the $M_2$ generations seedlings (35–40 plants for each family) were screened for their response to *Bgt* isolate E09 in a greenhouse. Susceptible $M_2$ plants were advanced to $M_3$ generation, which were inoculated again with *Bgt* isolate E09 to confirm their susceptibility. Twenty-six independent susceptible mutants derived from different $M_2$ families were identified and reconfirmed in the $M_3$ generation. The full-length genomic sequence of the *WTK3* and *CNL* genes from each of the 26 mutants were obtained using the primers listed in Supplementary Table 2. PCRs were performed in 20 μL volumes containing 10 μL 2 × Phanta Super-Fidelity DNA Polymeras (Vazyme, Nanjing, China), 20 ng of each primer, 100 ng genomic DNA. PCR conditions were initial denaturation at 95 °C for 3 min followed by 35 cycles of 95 °C for 15 s, 50–60 °C (depending on specific primers) for 15 s, and 72 °C for 30 s/kb, with a final extension at 72 °C for 5 min. The PCR products were sequenced by Sanger dye-terminator method. The sequences of the *WTK3* and *CNL* genes between the mutants and the wild-type HLT were compared using DNAMAN 8 software (Lynnon Biosoft, San Ramon, CA, USA).

**Genetic and physical mapping.** The *Ae. tauschii* genome sequence of the *MlHLT* region was used as templates for markers development (Supplementary Table 2). Genotyping of the fine mapping population and recombinants were also performed in 10 μL volumes with parameters as described above. PCR products were mixed with 2 μL loading buffer (98% formamide, 10 mM EDTA, 0.25% bromophenol blue, and 0.25% xylene cyanol) and separation in 8% non-denaturing polyacrylamide gels (39 acrylamide: 1 bisacrylamide). Gels were silver-stained and photographed. The linkage relationship of polymorphic markers and *MlHLT* was

established using Mapmaker 3.0[43], with the Kosambi map function and the logarithm of odd (LOD) score threshold was set at 3.0. Putative genes were annotated with the TriAnnot pipeline (https://urgi.versailles.inra.fr/triannot/?pipeline) and confirmed using the BLAST analysis tools available at NCBI (https://www.ncbi.nlm.nih.gov) and EnsemblPlants (http://plants.ensembl.org).

**cDNA analysis.** Total RNA was isolated from wheat leaf tissue inoculated with *Bgt* isolate E09 at 24 hpi, using the TRIzol reagent (Tiangen, Beijing, China). Reverse transcription was performed using a PrimeScript RT reagent Kit with gDNA Eraser (Takara, Kyoto, Japan). The amplification products from 5′ and 3′ RACE of the *WTK3* and *CNL* genes were cloned using the SMARTer RACE 5′/3′ Kit (Clontech, Kyoto, Japan). Forty colonies per reaction were selected for sequencing using the Sanger chain termination method. Cloning of the full-length *WTK3* cDNAs from HLT and S4185 were performed using Phanta Super-Fidelity DNA Polymeras (Vazyme, Nanjing, China) with the primers WTK3-CDSF/R, and 60 colonies per reaction were selected for Sanger sequencing.

**Plant genetic transformation.** The genomic DNA fragments of *WTK3* and *CNL* genes were obtained by PCR amplification from HLT. Restriction sites (*Bam*HI for *WTK3*, and *Sma*I and *Sbf*I for *CNL*) were added to the primers for cloning the PCR products.

The digested PCR products were cut by appropriate restriction enzymes, recovered and cloned from the 1% agarose gel, and cloned into the linearized pCAMBIA1300 vector via pEASY-Uni Seamless Cloning and Assembly Kit (Transgen Biotech, Beijing, China) to obtain the *ProWTK3:WTK3* and *ProCNL:CNL* plasmids. The cDNA of the *WTK3* main isoform (WTK3[HLT].IF1) was amplified from HLT using the primers WTK3-OETransF/R. The PCR products were cloned into the *Kpn*I-*Spe*I linearized pTCK303 vector to obtain the plasmid of *ProUbi:WTK3*. The three chimeric variants of WTK3 (WTK3#1, WTK3#2, and WTK3#3) were amplified from the *ProUbi:WTK3*. All the primers used are listed in Supplementary Table 2. The PCR products were cloned into the *Kpn*I-*Spe*I linearized pTCK303 vector to obtain the plasmids *ProUbi:WTK3#1*, *ProUbi:WTK3#2*, and *ProUbi:WTK3#3*, respectively. The colonies were sequenced to confirm the right DNA fragment insertions.

The five plasmids (*ProWTK3:WTK3*, *ProUbi:WTK3*, *ProUbi:WTK3#1*, *ProUbi:WTK3#2*, *ProUbi:WTK3#3*, and *ProCNL:CNL*) were transformed into the *Agrobacterium tumefaciens* strain EHA105 and delivered into the soft white spring wheat cultivar Fielder. *STS-Pm24* specific marker was used to detect the presence and expression of all the five versions of *WTK3* transgenes in the transgenic plants. PCR primer pair COM-CNLV-F/R was used to detect the presence of the transgene in the *CNL* transgenic plant. PCRs were performed in 20 μL volumes as described above. The expected sizes of the amplified PCR products of *STS-Pm24* in the wild-type Fielder was 95 bp for both DNA and cDNA templates. However, *STS-Pm24* amplified 89, 89, 92, 92, and 83 bp target fragments, in addition to the 95 bp Fielder fragment, in the positive transgenic plants of the five plasmids, respectively. Primer pair WTK3-CDSF/R was used to analyze the expression of the right ORFs of transgenes *WTK3*, *WTK3#1*, *WTK3#2*, and *WTK3#3* in the transgenic plants. Twenty $T_1$ transgenic plants (two-leaf stage) for each transgenic event were tested for their responses to the *Bgt* isolate E09, and disease symptoms were recorded 10 d post inoculation (dpi).

**Haplotype analysis.** Ten plants (two-leaf stage) of each germplasm used in the *WTK3* haplotype analysis were scored for their reactions to *Bgt* isolate E09 at 10 dpi using a 0–4 scale[26]. PCR primer pairs used for amplifying the entire *WTK3* gene and the 632 bp DNA fragment are listed in Supplementary Table 2. DNA amplifications were performed in a 20 μl reaction mixture containing 10 μl 2 × Phanta Super-Fidelity DNA Polymeras (Vazyme, Nanjing, China), 20 ng of each primer, and 100 ng of genomic DNA. DNA amplification was performed at 95 °C for 3 min, followed by 35 cycles at 95 °C for 15 s, 55–58 °C for 15 s depending on the annealing temperatures of primer pairs, and 72 °C for 30 s/kb, with a final extension at 72 °C for 5 min. All PCR products were sequenced using Sanger chain termination method, and the sequences were assembled using SeqMan in the LaserGene package. Sequences were analyzed using DNAMAN 8 software (Lynnon Biosoft, San Ramon, CA, USA) and aligned using cluster omega (https://www.ebi.ac.uk/Tools/msa/clustalo/). PCR products by the *Pm24* gene-specific STS marker *STS-Pm24* were mixed with 2 μl of loading buffer (98% formamide, 10 mM EDTA, 0.25% bromophenol blue, and 0.25% xylene cyanol), separated on 8% non-denaturing polyacrylamide gels (29 acrylamide: 1 bisacrylamide), and visualized after silver staining.

**Quantitative real-time RT-PCR.** Total RNA samples were extracted from two-leaf stage seedling leaves immediately before inoculation (0 h) and 4, 12, 24, 36, and 48 h post inoculation (hpi) with *Bgt* isolate E09 using the TRIzol reagent (Tiangen, Beijing, China). Quantitative real time RT-PCR (qPCR) was performed in a Roche 480 light cycler (Roche, Colorado Springs, CO, USA) and analyzed using a SYBR Premix Ex Taq II (TaKaRa, Kyoto, Japan). Primers used to evaluate the transcript levels of the *Pm24* candidate gene and the *PR* genes (*PR1*, *PR2*, *PR3*, *PR4*, *PR5*, and *PR9*) are listed in Supplementary Table S2. The wheat *ACTIN* gene was used as the

endogenous control. The comparative threshold $2^{-\Delta\Delta CT}$ method was used to quantify relative gene expression[55]. Each sample was analyzed with three replicates.

**Phylogenetic analysis of plant protein kinase domains**. The 174 putative kinase or pseudokinase domains used for phylogenetic analysis of WTK1 (Yr15)[35] and WTK2 (Sr60)[36] were also used in this study. The other 8 proteins homologous to WTK3 with a TKP structure were retrieved from whole-genome assemblies of various plant taxa: T. aestivum (http://202.194.139.32/blast/viroblast.php), H. vulgare (http://webblast.ipk-gatersleben.de/barley_ibsc/), B. distachyon (https://phytozome.jgi.doe.gov/pz/portal.html), O. sativa, Z. mays, and S. bicolor (https://www.ncbi.nlm.nih.gov/). Conserved kinase domains were identified using SMART[56] (http://smart.embl-heidelberg.de/smart/set_mode.cgi?NORMAL = 1). A phylogenetic tree (Neighbour-joining tree) was computed with Clustal Omega[57] (https://www.ebi.ac.uk/Tools/msa/clustalo/) and drawn with iTOL (https://itol.embl.de/).

**Reporting summary**. Further information on research design is available in the Nature Research Reporting Summary linked to this article.

## Data availability

Data supporting the findings of this work are available within the paper and its Supplementary Information files. The datasets generated and analyzed during the current study are available from the corresponding author upon request. Detailed sequence data of Pm24 can be found in National Center for Biotechnology Information (NCBI) under the accession number MK950855. The source data underlying Figs. 1a, 2b, c, 3b as well as Supplementary Figs. 1 and 2 are provided as a Source Data file.

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

## Acknowledgements

We are grateful to Prof. Qixin Sun and Tsomin Yang from China Agricultural University, Beijing, China, for their advices and supports during this research. Many thanks for Drs. Xueyong Zhang, Xianchun Xia, and Lihui Li of Chinese Academy of Agricultural Sciences, Beijing, China for providing the Chinese wheat mini-core collection (MCC), worldwide wheat landrace collection, and Chinese wheat landraces, respectively; Drs. Jirui Wang of Sichuan Agricultural University, Chengdu, China, and Lingrang Kong of Shandong Agricultural University, Tai'an, China, for providing *Ae. tauschii* accessions; Prof. Wanquan Ji of Northwest A & F University, Yangling, China, for the gift seeds of landrace BHL; and Dr. Shisheng Chen of Peking University Institute of Advanced Agricultural Sciences, Weifang, Shandong, China, for providing TKP sequences for phylogenetic analysis. We are also grateful to Dr. Genying Li of Shandong Academy of Agriculture Sciences, Jinan, China, for developing the transgenic wheat. We gratefully appreciate Dr. Robert L. Conner, Morden Research and Development Centre, Agriculture and Agri-Food Canada, Morden, Manitoba, Canada, for careful improvement of the manuscript. This research was financially supported by the Strategic Priority Research Program of the Chinese Academy of Sciences (XDA24010305), the National Key Research and Development Program of China (2016YFD0100302), the National Natural Science Foundation of China (31801345 to P.L.; 30425039 and 31030056 to Z.L.), Science and Technology Service Network Initiative of Chinese Academy of Sciences (KFJ-STS-ZDTP-024 to Z.L.) and the National Science Foundation (NSF) of the United States under grant number IOS-1238231.

## Author contributions

Z.L. and H.L. designed the study. P.L., L.G., Z.W., B.L., J.L., Y.L., D.Q., W.S., L.Y., N.W., G.G., Q.W., Y.C., M.L., H.Z., P.Z., K.Z., Y.Z., K.R.D. and N.H. performed the research. J.X., L.D., C.L., D.Y., M.C.L., J.D. and Y.Q.G. analyzed the data. P.L., H.L and Z.L. wrote the paper.

## Competing interests

The authors declare no competing interests.
