## [Peer Review File · Nature Communications]

Reviewers' comments:

Reviewer #1 (Remarks to the Author):

Ping Lu and colleagues present a solid piece of work on the identification (cloning) and characterisation of the powdery mildew resistance gene Pm24 from hexaploid bread wheat. Their findings present new and exciting evidence on the involvement of tandem kinases in disease resistance. Previous studies have shown the involvement of a tandem kinase to stem rust resistance in barley (Rpg1) and, more recently, stripe rust resistance in wheat (Yr15). This is the first report of a tandem kinase resistance gene effective against powdery mildew. The identification of Pm24 is backed up by (i) bi-parental map-based approximation (restricting the gene to a discrete ~500 kb interval), (ii) extensive and fully convincing transgenics, and (iii) EMS mutagenesis, screening and sequencing of susceptible mutants. The authors thus show that the identified gene is both sufficient and required for disease resistance. The paper is succinct and to the point, very well written, and the figures are well designed, easy to follow, nicely synchronised with the main text, and in the majority of the cases underpinned by the statements given in the text. I expect this paper will be of great interest to the community working on cereal disease resistance and also gain some traction with the broader molecular plant-microbe interaction community.

I have only some minor concerns, which I think can be largely dealt with by rephrasing text and/or adding a qualifier to a few of the statements (except for point 4 below):

1. Lines 41, 154, and 612. Strictly speaking I think the use of the word "complementation" is incorrect here. A mutant has not been 'complemented'. Rather, resistance has been 'conferred' upon the otherwise powdery mildew susceptible cultivar Fielder. The authors might consider rephrasing the text to reflect this distinction.

2. Lines 76 to 78 in the introduction build up some of the motivation for cloning Pm24 and state that: "broad-spectrum resistance genes are usually more robust and do not fast succumb to emergence of new virulent [pathogen isolates]". This is, at best, a risky statement to make. First of all, there is little evidence to back up the statement and it does not feel right to back it up by referring to the pathogen-non-specific and partial resistance genes Lr34 and Lr67. There are likely many major broad-spectrum resistance genes in wild relatives of our crop plants that, were they deployed on a large scale in modern mono-culture cropping systems, would succumb very rapidly. I suggest the authors rephrase this section to tone down the statement or provide qualifiers. Also, the grammar in lines 77-78 needs fixing.

3. Line 242: Please provide a supplementary Excel sheet with a list of the 1,069 common wheat and *Ae. tauschii* accessions tested and their provenance.

4. Figure 4d shows that Pm24 fused to GFP and transiently overexpressed in wheat protoplasts accumulates in the nucleus and cytoplasm. The protein accumulation pattern appears to be very similar to that of the GFP only control construct. It would be reassuring therefore if the authors could provide a gel to show that the Pm24:GFP construct remains intact during the transient expression.

5. Line 299 (and elsewhere): The authors state that Pm24 is a broad-spectrum resistance gene based on screening against 93 Bgt isolates and not discovering any strains with virulence on Pm24. The strains were collected from geographically distinct regions of China. However, some evidence is required to exclude the formal possibility that the 93 Bgt isolates are largely clonal – in the absence of this, the claim of broad-spectrum resistance is invalid. Providing these isolates have previously been characterised and fulfil the criteria of being genetically diverse and representative of the species-wide genetic diversity, then a citation of this work will suffice. In the absence of this, I suggest toning down or adding a qualifier to the statements of broad-spectrum resistance,

e.g. the authors might consider settling for 'confers resistance to prevalent Bgt isolates across China' or the alike.

6. An important resource output from this paper are a number of constructs for Pm24. It would be nice for many obvious reasons if the authors deposited these constructs in the public repository Addgene to ensure that they are freely available to the scientific community.

Other minor issues/typos that I noted are as follows:

1. Line 58: I suggest rephrasing: "Diseases and pests are serious threat to the production of wheat grain..." to "Diseases and pests pose a serious threat to the production of wheat grain..."
2. Line 114: I suggest rephrasing: "mapped into..." to "mapped to..."
3. Line 156: Remove surplus "and" in sentence.
4. Line 206: Consider removing the word "examined" to improve flow of sentence.
5. Line 275: Should it not be "regulate"?
6. Line 280: The authors state here that Pm24 encodes an "active serine/threonine kinase". This statement would require protein biochemistry to back up. As this was not part of the present study I suggest that the authors rephrase this statement.
7. Line 601 and 602: "um" should be "µm".
8. Line 642: Please fix grammar.
9. Line 674: "similar with" should be "similar to"
10. Line 681: The plural of rhombus is rhombi or rhombuses.
11. Line 706: "Aegilops" should all be in italics.

Reviewer #2 (Remarks to the Author):

The authors described the map-based cloning of the Pm24 gene that confers resistance against powdery mildew in wheat. The work including transgenics and mutant look rigorous. There is no doubt about the candidate. Pm24 encodes a protein with two kinases domains. This reinforces the fact that this gene family is an important player in disease resistance and would require more attention. This work is quite original compared to the current literature in a way that there are very few examples (if any) showing that a specific 6 bp deletion confers resistance. In addition, I find quite interesting the different subcellular localization of the resistant haplotype despite it would require more rigorous experiments. This gives us some clues about the potential role of this 6 bp deletion and how the resistance is initiated.

There are a numbers of issues that need to be corrected in the manuscript:

1/Throughout the manuscript, the authors insist on the broad-spectrum resistance conferred by Pm24. There are two points that bother me:

- The authors can't say in the discussion that "Pm24 that confers resistance to all of the 93 bgt isolates (L300)". This is not true as there might be other genes that control resistance to these isolates in the wheat carrying Pm24. If they would like to tell that Pm24 confers resistance against these isolates it would require evaluating their transgenics (positive and negative sister lines) against these 93 isolates and not only against one isolate E09. I suggest testing their transgenic with a certain number of isolates to provide evidences that it confers broad-spectrum resistance.
- In the introduction, the authors describe two examples of broad-spectrum resistance but both confer partial resistance. In the case of Pm24 it is a strong qualitative resistance. This means that this gene would be one of the first examples of strong broad-spectrum resistance against powdery mildew? A gene that was never exposed over large area to the pathogen could explain its broad-spectrum resistance. I suggest to genotype a collection of modern wheat varieties (including worldwide accessions) with their diagnostic STS marker. This will let them know if the gene is already present in the cultivated wheat and has been exposed to the pathogen.

-

2/As there are very few information about how the WTK#1, 2, 3 constructs were done. We may argue that the process used to produce deleted copies of the gene (3bp or 12bp) could have led to nonfunctional gene copies as the 6-bp deletion was not performed the same way.

3/I would recommend making in silico 3D structure of the protein with or without the deletion in order to see the impact of the deletion on protein structure.

4/Regarding the subcellular localization, there is no western blot indicating that the WKSHLT-GFP is not cleaved and the signal observed in the nucleus not due to free GFP. In addition, as the difference in subcellular localization is quite original I would recommend performing additional experiment in another in vivo system like tobacco.

5/Regarding paragraph about Pm24 evolution, are the proteins identified the true orthologous. This is a large family. How the search for orthologous was performed?

6/I'm a bit disappointed by the discussion. There is clearly a lack of comparison with other well-known cytoplasmic kinases like BIK1 or Pto.

7/There is clearly a lack of details in the Material and Methods section that need to be addressed (Please see below).

Additional points:

There is a great recent review about worldwide yield losses caused by devastating pathogens (Savary et al. 2019. I think it is worth citing it.

L69 please say "cloned and characterized"

Please explain what kinds of markers are WGG? SNPs, SSRs?

L156, why are the constructs complementary?

Sometimes authors included (as exponent) the genotype origin in their construct names. This makes things much better to understand and I would recommend following this nomenclature for all construct names. (especially in the transgenic complementation paragraph and Figure 2).

Data of only three T1 families are presented in the document. Could authors give the scores in a supplementary table for all families?

Could authors provide the sequence of the different isoforms.

L177: According to the SM3, the alternative transcripts between S4185 and HLT are not similar. Please correct.

The authors identified 26 susceptible families and only eleven have a mutation in WKS, what about the 15 others?

Could authors indicate the score on Figure 2 and SM table 3 for the mutants as it seems that the level of susceptibility is different?

L197: 477 or 478?

L222 representative of what ?

Please provide the names of the 20 accessions used for WTK re-sequencing.

L240 Authors said that accession Hongmangmai carries as well the 6-bp deletion and it is highly resistant to powdery mildew. No phenotypic evaluation of this accession is included in the manuscript. I would appreciate to see data against a set of isolates (if this is not already done) as well as the entire Pm24 CDS sequence of this accession (to make sure it share the same haplotype with the three other resistant).

The Mat and Meth section is very poor. There is a lot of missing information. In addition, order of methods in the Mat and Meth section is different from how they occur in the main text. So please correct this.

Missing information:

L379 "with some modifications" Which ones ? Please indicate how many plants have been scored

per isolate

L410 PCR condition for WTK gene re-sequencing.

L415 please indicate a reference

How genotyping assays were performed?

Which Chinese spring genome sequence

L419-420: a set of different tools were used for annotation but how annotations from different tools were merged?

L427/428 please indicate the primers used.

L431 PCR amplification, please detail.

There is no explanation of how genes candidate expression was performed. How they have been cloned and sequenced ?

How many plants per T1 family have been evaluated?

Nothing about re-sequencing (PCR condition) and haplotype comparison.

Could authors indicate the specific condition used to amplify the STS marker?

Figure 1:

The title is Map-based cloning of Pm24. In the main text it is referred as Map-based cloning of MIHLT. At this step authors are not 100% sure that both genes are allelic I would keep MIHLT in Figure 1 as well. Remove Pm24 from the legend and change by MIHLT.

N=3720, it can be confusing between number of gametes and real number of individuals, I would replace it by 3,720 F2

The scales used for c, e and f are not exact according to the distance indicated in the main text.

There is a gap between scaffold 3178.1 and scaffold3268.1 that you do not indicate in e. What does this mean?

In the legend, please indicate the mapping population used for genetic mapping, the origin of the physical map and which Chinese spring genome sequence was used. Please explain what are RLK, WTK, HP...

Figure 2

Same remark as Figure 1 for the title same for SM Figure2 Figure4

I would add "genomic or native" sequence of the WTK gene.

Above the pictures is written CNL-CPL1. and in the text CPL-CNL It is confusing. Please correct.

Figure 3

Could authors indicate which accessions belong to the different categories (common wheat landraces, mini core collection...)

How was done the clustering of haplotypes?

In "a" please indicate where is the 6-bp deletion located.

Please clarify the legends about the position of SNP, deletion and insertion. Which accession is the reference?

I'm surprised to see that HLT and Shi4185 belong to the same haplotypic group. Can you explain?

What represent the nucleotides in red? Please indicate in the legend.

Figure 4

There is no LB and RB in the figure compared to the legend.

SM Figure 2

There is no information about the number of replicates. Please perform statistical analysis to see if the differences in expression are significant along the time course.

SM3

Please indicates what the numbers in brackets are.

There is a typo in the legend.

Please indicate clearly in the legend the differences between the variants, e.g. fourth intron not spliced....

SM figure 4

There is a shift between the names of the sequences and the sequences which makes the figure difficult to read.

I'm not quite sure to understand why you said that "Pm24 protein structure was similar with Yr15". Here first you look at the protein sequence and it is therefore difficult to talk about structure and I observed many differences between both sequences.

Please explain "KinI"

I think it would have been better to separate domain I and domain II in two different alignments. Mut614 G718R is located on a position where there is no G. Please correct. Same for Mut477

SM Table 4

Please provide information about the isolate used to obtain the IF. What does CWMCC mean?

The title mentioned distribution of haplotypes but there is no information about haplotypes.

Dear Reviewers,

Thank you so much for reviewing our manuscript and we appreciate the detailed comments and suggestions in helping us to improve the manuscript. We have revised the manuscript based on those comments and suggestions. Enclosed please find our revised manuscript and the responses to the raised questions.

Thank you again for your kind comments!

Reviewer comments

Reviewer #1 (Remarks to the Author):

Ping Lu and colleagues present a solid piece of work on the identification (cloning) and characterisation of the powdery mildew resistance gene Pm24 from hexaploid bread wheat. Their findings present new and exciting evidence on the involvement of tandem kinases in disease resistance. Previous studies have shown the involvement of a tandem kinase to stem rust resistance in barley (Rpg1) and, more recently, stripe rust resistance in wheat (Yr15). This is the first report of a tandem kinase resistance gene effective against powdery mildew. The identification of Pm24 is backed up by (i) bi-parental map-based approximation (restricting the gene to a discrete ~500 kb interval), (ii) extensive and fully convincing transgenics, and (iii) EMS mutagenesis, screening and sequencing of susceptible mutants. The authors thus show that the identified gene is both sufficient and required for disease resistance. The paper is succinct and to the point, very well written, and the figures are well designed, easy to follow, nicely synchronised with the main text, and in the majority of the cases underpinned by the statements given in the text. I expect this paper will be of great interest to the community working on cereal disease resistance and also gain some traction with the broader molecular plant-microbe interaction community.

I have only some minor concerns, which I think can be largely dealt with by rephrasing text and/or adding a qualifier to a few of the statements (except for point 4 below):

1. Lines 41, 154, and 612. Strictly speaking I think the use of the word “complementation” is incorrect here. A mutant has not been ‘complemented’. Rather, resistance has been ‘conferred’ upon the otherwise powdery mildew susceptible cultivar Fielder. The authors might consider rephrasing the text to

reflect this distinction.

Reply: *Thanks for the suggestions. The transgenic wheat driven by the native promoter was rephrased to replace “complementation” in the revised manuscript.*

2. Lines 76 to 78 in the introduction build up some of the motivation for cloning Pm24 and state that: “broad-spectrum resistance genes are usually more robust and do not fast succumb to emergence of new virulent [pathogen isolates]”. This is, at best, a risky statement to make. First of all, there is little evidence to back up the statement and it does not feel right to back it up by referring to the pathogen-non-specific and partial resistance genes Lr34 and Lr67. There are likely many major broad-spectrum resistance genes in wild relatives of our crop plants that, were they deployed on a large scale in modern mono-culture cropping systems, would succumb very rapidly. I suggest the authors rephrase this section to tone down the statement or provide qualifiers. Also, the grammar in lines 77-78 needs fixing.

Reply: *The paragraph was revised as suggested to avoid over-statement of the “spectrum” of the Pm24 gene.*

3. Line 242: Please provide a supplementary Excel sheet with a list of the 1,069 common wheat and *Ae. tauschii* accessions tested and their provenance.

Reply: *The full list of the 1,069 common wheat and *Aegilops tauschii* accessions tested for reaction to Bgt isolate E09 and haplotype analysis was provided as Supplementary Table 5.*

4. Figure 4d shows that Pm24 fused to GFP and transiently overexpressed in wheat protoplasts accumulates in the nucleus and cytoplasm. The protein accumulation pattern appears to be very similar to that of the GFP only control construct. It would be reassuring therefore if the authors could provide a gel to show that the Pm24:GFP construct remains intact during the transient expression.

Reply: *Many thanks for the suggestion. We performed subcellular localizations using *Nicotiana benthamiana* agro-infiltration assay. The green fluorescent protein (GFP), and fusion proteins of GFP with WTK3^{HLT}, WTK3^{S4185}, WTK3#1, WTK3#2 and WTK3#3 constructs [pCAMBIA1300GFP (control GFP), Pro35S:WTK3^{HLT}•GFP, Pro35S:WTK3^{S4185}•GFP, Pro35S:WTK3#1•GFP, Pro35S:WTK3#2•GFP and Pro35S:WTK3#3•GFP] under the cauliflower mosaic virus 35S promoter (35S) were used to transform *Agrobacterium tumefaciens* strain GV3101. The young apical leaves (5–6 leaves per plant) of *N. benthamiana* were infiltrated with recombinant *Agrobacterium* strains using a syringe (2 mL) without needle (Yuan et al. 2011). The agroinfiltrated leaves were photographed 36 h following infiltration. We found that the WTK3^{HLT}•GFP, WTK3^{S4185}•GFP, WTK3#1•GFP, WTK3#2•GFP and WTK3#3•GFP fused proteins localized in both the cytoplasm and nucleus.*

We then repeat the subcellular localizations of the green fluorescent protein (GFP), and fusion proteins of GFP with $WTK3^{HLT}$, $WTK3^{S4185}$, $WTK3\#1$, $WTK3\#2$ and $WTK3\#3$ constructs using wheat protoplast assay (Wang et al. 2014). The GFP and fusion proteins constructs under the promoter of *Z. maize* ubiquitin gene [pJIT163GFP (control GFP), ProUbi: $WTK3^{HLT}$ •GFP, ProUbi: $WTK3^{S4185}$ •GFP, ProUbi: $WTK3\#1$ •GFP, ProUbi: $WTK3\#2$ •GFP and ProUbi: $WTK3\#3$ •GFP (fused GFP)] were introduced into wheat protoplasts (Xiao et al. 2014). The protoplasts were photographed at 16 h and 36 h after transformation. We found that $WTK3^{HLT}$ GFP is localized in the cytoplasm with limited presence in the nucleus, but the fusion proteins of GFP with $WTK3$ susceptible allele variants ($WTK3^{S4185}$ GFP, $WTK3\#1$ GFP, $WTK3\#2$ GFP, and $WTK3\#3$ GFP) are detectable almost in the cytoplasm (Figure below in response to reviewer 2). However, when photographed at 36 h after transformation, we found that the fusion proteins of $WTK3^{HLT}$ GFP, $WTK3^{S4185}$ GFP, $WTK3\#1$ GFP, $WTK3\#2$ GFP and $WTK3\#3$ GFP can be detected in both cytoplasm and nucleus.

The reason for the difference of the subcellular locations of the $WTK3$ and its variants in wheat protoplasts at 16 h and 36 h after transformation could not be explained right now. In order to have a rigorous and more reliable result of the relationship between the subcellular localizations of the $WTK3$ and its susceptible variants to the powdery mildew resistance, we are developing stable transgenic wheat of the $WTK3^{HLT}$ with a nuclear export signal (NES) and $WTK3^{S4185}$ with a nuclear localization signal (NLS). Since we need extra time to generate the transgenic wheat and test their reactions to powdery mildew pathogens, we would like to not include the subcellular localization results in the current version of the manuscript and leave it to future study.

5. Line 299 (and elsewhere): The authors state that Pm24 is a broad-spectrum resistance gene based on screening against 93 Bgt isolates and not discovering any strains with virulence on Pm24. The strains were collected from geographically distinct regions of China. However, some evidence is required to exclude the formal possibility that the 93 Bgt isolates are largely clonal – in the absence of this, the claim of broad-spectrum resistance is invalid. Providing these isolates have previously been characterised and fulfil the criteria of being genetically diverse and representative of the species-wide genetic diversity, then a citation of this work will suffice. In the absence of this, I suggest toning down or adding a qualifier to the statements of broad-spectrum resistance, e.g. the authors might consider settling for ‘confers resistance to prevalent Bgt isolates across China’ or the alike.

Reply: Revised as suggested to avoid statement of “broad-spectrum” resistance. The 93 Bgt isolates were re-sequenced genetically divergent Bgt collections in China and used for genome wide association study of AvrPm3

(McNally et al. *New Phytol.* 2018, 218:681-695). A citation of this publication (Ref. 27) was provided in the revised manuscript.

6. An important resource output from this paper are a number of constructs for Pm24. It would be nice for many obvious reasons if the authors deposited these constructs in the public repository Addgene to ensure that they are freely available to the scientific community.

Reply: *This study was supported by the public funds and the constructs of the WTK3 are freely available up request to the corresponding authors.*

Other minor issues/typos that I noted are as follows:

1. Line 58: I suggest rephrasing: “Diseases and pests are serious threat to the production of wheat grain...” to “Diseases and pests pose a serious threat to the production of wheat grain...”

Reply: *This has been changed accordingly.*

2. Line 114: I suggest rephrasing: “mapped into...” to “mapped to...”.

Reply: *This has been changed accordingly.*

3. Line 156: Remove surplus “and” in sentence.

Reply: *This has been changed accordingly.*

4. Line 206: Consider removing the word “examined” to improve flow of sentence.

Reply: *This has been changed accordingly.*

5. Line 275: Should it not be “regulate”?

Reply: *This has been changed accordingly.*

6. Line 280: The authors state here that Pm24 encodes an “active serine/threonine kinase”. This statement would require protein biochemistry to back up. As this was not part of the present study I suggest that the authors rephrase this statement.

Reply: *This has been changed accordingly to “putative tandem kinase protein” in the revised manuscript. The kinase activity will be characterized in future study.*

7. Line 601 and 602: “um” should be “µm”.

Reply: *This has been changed accordingly.*

8. Line 642: Please fix grammar.

Reply: *This has been corrected.*

9. Line 674: “similar with” should be “similar to”

Reply: *This has been changed accordingly.*

10. Line 681: The plural of rhombus is rhombi or rhombuses.

Reply: *This has been changed accordingly.*

11. Line 706: “Aegilops” should all be in italics.

Reply: *This has been changed accordingly.*

Reviewer #2 (Remarks to the Author):

The authors described the map-based cloning of the Pm24 gene that confers resistance against powdery mildew in wheat. The work including transgenics and mutant look rigorous. There is no doubt about the candidate. Pm24 encodes a protein with two kinases domains. This reinforces the fact that this gene family is an important player in disease resistance and would require more attention. This work is quite original compared to the current literature in a way that there are very few examples (if any) showing that a specific 6 bp deletion confers resistance. In addition, I find quite interesting the different subcellular localization of the resistant haplotype despite it would require more rigorous experiments. This gives us some clues about the potential role of this 6 bp deletion and how the resistance is initiated.

There are a numbers of issues that need to be corrected in the manuscript:

1/Throughout the manuscript, the authors insist on the broad-spectrum resistance conferred by Pm24. There are two points that bother me:

- The authors can't say in the discussion that "Pm24 that confers resistance to all of the 93 bgt isolates (L300)". This is not true as there might be other genes that control resistance to these isolates in the wheat carrying Pm24. If they would like to tell that Pm24 confers resistance against these isolates it would require evaluating their transgenics (positive and negative sister lines) against these 93 isolates and not only against one isolate E09. I suggest testing their transgenic with a certain number of isolates to provide evidences that it confers broad-spectrum resistance.

Reply: *Many thanks for the suggestions. In the revised manuscript, we rephrased the statement of "broad-spectrum". The 93 Bgt isolates were re-sequenced genetically divergent Bgt collections in China and used for genome-wide association study of AvrPm3 (McNally et al. 2018 New Phytol. 218:681-695). A citation of this publication (Ref. 27) was provided in the revised manuscript. Since we need more phenotypic data for the Pm24 gene to more diversified Bgt isolates inoculation, we just mentioned "In the current study, we showed that Pm24 confers resistance to all of the available 93 divergent Bgt isolates collected from 12 provinces of China, suggesting that WTK3 may also be a putative broad-spectrum resistance protein."*

- In the introduction, the authors describe two examples of broad-spectrum resistance but both confer partial resistance. In the case of Pm24 it is a strong qualitative resistance. This means that this gene would be one of the first examples of strong broad-spectrum resistance against powdery mildew? A gene that was never exposed over large area to the pathogen could explain its

broad-spectrum resistance. I suggest to genotype a collection of modern wheat varieties (including worldwide accessions) with their diagnostic STS marker. This will let them know if the gene is already present in the cultivated wheat and has been exposed to the pathogen.

Reply: *Many thanks for the suggestions. We tested a collection of 1,069 wheat and *Aegilops tauschii* accessions collected from geographically diverse regions worldwide using the diagnostic STS marker to reveal the presence of the 6-bp deletion. A supplementary Table 5 was provided for detail information of the results. Our result indicated that the Pm24 allele may originate recently after common wheat was introduced into central China as a natural mutation that has not widely spread into other landraces and not used in modern wheat breeding program.*

-

2/As there are very few information about how the WTK#1, 2, 3 constructs were done. We may argue that the process used to produce deleted copies of the gene (3bp or 12bp) could have led to nonfunctional gene copies as the 6-bp deletion was not performed the same way.

Reply: *Detail information of the chimeric WTK3 construct variants was provided in the revised manuscript. The full-length cDNAs of the WTK3 variants in the transgenic wheat were also obtained to confirm the right transcripts of each constructs and ORF for functional transgene copies in the positive transgenic plants.*

3/I would recommend making in silico 3D structure of the protein with or without the deletion in order to see the impact of the deletion on protein structure.

Reply: *The 2D and 3D structures of the WTK3 protein were predicted using PSIPRED (<http://bioinf.cs.ucl.ac.uk/psipred/>), and the RosettaCM method. The result was provided as Supplementary Figure 10 in the revised manuscript. The 6-bp deletion (K399G400) was just located between two predicted α -helixes (383-397, and 405-422) in WTK3. The 6-bp deletion shortens the interval loop between the two α -helixes. The information was updated in the revised manuscript.*

4/Regarding the subcellular localization, there is no western blot indicating that the WKSHLT-GFP is not cleaved and the signal observed in the nucleus not due to free GFP. In addition, as the difference in subcellular localization is quite original I would recommend performing additional experiment in another in vivo system like tobacco.

Reply: *Many thanks for the suggestion. We performed subcellular localizations using *Nicotiana benthamiana* agro-infiltration assay. The green fluorescent protein (GFP), and fusion proteins of GFP with WTK3^{HLT}, WTK3^{S4185}, WTK3#1, WTK3#2 and WTK3#3 constructs [pCAMBIA1300GFP (control GFP), Pro35S:WTK3^{HLT}•GFP, Pro35S:WTK3^{S4185}•GFP,*

Pro35S:WTK3#1•GFP, Pro35S:WTK3#2•GFP and Pro35S:WTK3#3•GFP under the cauliflower mosaic virus 35S promoter (35S) were used to transform *Agrobacterium tumefaciens* strain GV3101. The young apical leaves (5–6 leaves per plant) of *N. benthamiana* were infiltrated with recombinant *Agrobacterium* strains using a syringe (2 mL) without needle (Yuan et al. 2011). The agro-infiltrated leaves were photographed 36 h following infiltration. We found that the *WTK3^{HLT}•GFP, WTK3^{S4185}•GFP, WTK3#1•GFP, WTK3#2•GFP* and *WTK3#3•GFP* fused proteins localized in both the cytoplasm and nucleus (Figure below).

We then repeat the subcellular localizations of the green fluorescent protein (GFP), and fusion proteins of GFP with *WTK3^{HLT}, WTK3^{S4185}, WTK3#1, WTK3#2* and *WTK3#3* constructs using wheat protoplast assay (Wang et al. 2014). The GFP and fusion proteins constructs under the promoter of *Z. maize* ubiquitin gene [*pJIT163GFP* (control GFP), *ProUbi:WTK3^{HLT}•GFP, ProUbi:WTK3^{S4185}•GFP, ProUbi:WTK3#1•GFP, ProUbi:WTK3#2•GFP* and *ProUbi:WTK3#3•GFP* (fused GFP)] were introduced into wheat protoplasts (Xiao et al. 2014). The protoplasts were photographed at 16 h and 36 h after transformation. We found that *WTK3^{HLT}GFP* is localized in the cytoplasm with limited presence in the nucleus, but the fusion proteins of GFP with *WTK3* susceptible allele variants (*WTK3^{S4185}GFP, WTK3#1GFP, WTK3#2GFP, and WTK3#3GFP*) are detectable almost in the cytoplasm (Figure below) at 16 hours after transformation. However, when photographed at 36h after transformation, we found that the fusion proteins of *WTK3^{HLT}GFP, WTK3^{S4185}GFP, WTK3#1GFP, WTK3#2GFP* and *WTK3#3GFP* can be detected in both cytoplasm and nucleus (Figure below).

The reason for the difference of the subcellular locations of the *WTK3* and its variants in wheat protoplasts at 16 h and 36 h after transformation could not be explained right now. In order to have a rigorous and more reliable result of the relationship between the subcellular localizations of the *WTK3* and its susceptible variants to the powdery mildew resistance, we are developing stable transgenic wheat of the *WTK3^{HLT}* with a nuclear export signal (NES) and *WTK3^{S4185}* with a nuclear localization signal (NLS). Since we need extra time to generate the transgenic wheat and test their reactions to powdery mildew pathogens, we would like to not include the subcellular localization results in the current version of the manuscript and leave to future study.

Reference:

Yuan et al. (2011) A high throughput Barley stripe mosaic virus vector for virus induced gene silencing in monocots and dicots. *PLoS ONE* 6:e26468
Wang, Y. P. et al. (2014) Simultaneous editing of three homoeoalleles in hexaploid bread wheat confers heritable resistance to powdery mildew. *Nature Biotechnology*. 32:947-951.

Xiao J, et al. (2014) O-GlcNAc-mediated interaction between VER2 and TaGRP2 elicits TaVRN1 mRNA accumulation during vernalization in winter wheat. *Nature Communications* 5:4572.

Figure Subcellular localization of WTK3^{HLT}, WTK3^{S4185}, WTK3#1, WTK3#2, and WTK3#3 in wheat protoplast and *Nicotiana benthamiana* leaves. a, ProUbi:WTK3^{HLT}GFP, ProUbi:WTK3^{S4185}GFP, ProUbi:WTK3#1GFP, ProUbi:WTK3#2GFP and ProUbi:WTK3#3GFP were introduced into wheat protoplast and photographed at the 36h after transformation. **b,** Pro35S:WTK3^{HLT}GFP, Pro35S:WTK3^{S4185}GFP, Pro35S:WTK3#1GFP, Pro35S:WTK3#2GFP and Pro35S:WTK3#3GFP were introduced into *Nicotiana benthamiana* leaves and photographed at the 48 h following agro-infiltration.

5/Regarding paragraph about Pm24 evolution, are the proteins identified the true orthologous. This is a large family. How the search for orthologous was performed?

Reply: we performed BLAST search of the available genome assemblies of Chinese spring (IWGSC Ver. 1.0), *Aegilops tauschii* (Ref. 28), durum wheat Svevo (Ref. 41), wild emmer Zavitan (Ref. 42), *Triticum urartu* (Ref. 43), barley

(Ref. 44) and rye (Ref. 45) to identify the highly homologous annotated genes, especially on the homoeologous chromosomes. The genomic regions of the WTK3 homologs were also analyzed to confirm the collinearity relationship (including the WTK3 and nearby genes) between the homoeologous chromosomes. Phylogenetic analysis was also performed to identify the true orthologous (Supplementary Figure 9).

6/I'm a bit disappointed by the discussion. There is clearly a lack of comparison with other well-known cytoplasmic kinases like BIK1 or Pto.

Reply: *The discussion was re-written and the information of several RLCKs was included in the revised manuscript. The kinase domains of BIK1, Pto, BAK1, BSK1, Stpk-V, and BIR1, etc. were used for kinase domain alignments to predict the potential kinase function of Kin I and Kin II of WTK3 (Pm24). The Kin I of WTK3 was predicted as a putative kinase and the Kin II of WTK3 was predicted as a pseudokinase (Supplementary Figure 4). The predicted 2D and 3D structures of WTK3 were also compared with BIK1 (Ref. 49). The information was included in the revised manuscript.*

7/There is clearly a lack of details in the Material and Methods section that need to be addressed (Please see below).

Reply: *More details were provided in the revised Material and Methods section to provide more information related to the performed research.*

Additional points:

There is a great recent review about worldwide yield losses caused by devastating pathogens (Savary et al. 2019). I think it is worth citing it.

Reply: *Revised as suggested.*

L69 please say "cloned and characterized"

Reply: *Revised as suggested.*

Please explain what kinds of markers are WGG? SNPs, SSRs?

Reply: *The full expression of single nucleotide polymorphisms (SNPs) and simple sequence repeats (SSRs) were revised as suggested. Molecular markers WGGBxxx and WGGCxxx were named after our lab initials in an order of developing time.*

L156, why are the constructs complementary?

Reply: *Sorry for the imprecise expression! The constructs with the native promoters of the transgenes were treated as "complementary constructs" for transgenic function validation. The sentence was revised to express the right meaning.*

Sometimes authors included (as exponent) the genotype origin in their construct names. This makes things much better to understand and I would recommend following this nomenclature for all construct names. (especially in the transgenic complementation paragraph and Figure 2).

Reply: *All of the constructs used in the current study were re-named following the common used nomenclature method as suggested.*

Data of only three T1 families are presented in the document. Could authors give the scores in a supplementary table for all families?

Reply: *The inoculation data of all the transgenic families was provided in a new supplementary Table 3.*

Could authors provide the sequence of the different isoforms.

Reply: *The genomic DNA and cDNA sequences of the different isoforms of WTK3^{HLT} and WTK3^{S4185} were provided in the revised Supplementary Figure 3.*

L177: According to the SM3, the alternative transcripts between S4185 and HLT are not similar. Please correct.

Reply: *Revised as suggested. The right information was provided in the revised Supplementary Figure 3.*

The authors identified 26 susceptible families and only eleven have a mutation in WKS, what about the 15 others?

Reply: *We didn't find any sequence variation in the other 15 susceptible EMS mutants on the entire WTK3 gene (including the promoter and downstream sequences). We speculate that these mutations may have sequence variations on other genes in the WTK3 regulation pathway, which we are making crosses for genetic analysis and gene mapping now. The information was included in the revised manuscript as "No sequence variation was detected for another 15 EMS mutants in the WTK3 and CNL genes (including putative promoters, exons and introns, and terminator regions), suggesting possible mutations on other unknown genes or elements involved in the Pm24 regulation pathway."*

Could authors indicate the score on Figure 2 and SM table 3 for the mutants as it seems that the level of susceptibility is different?

Reply: *The infection type (IT) of the mutants were classified as 4 (highly susceptible) according to a 0, 0;, 1 to 4 scales, representing immune, necrosis flecks, highly resistant, moderate resistant, moderate susceptible, and highly susceptible. Minor different between 3 and 4 were observed for some EMS mutants. However, based on the M₃ families IT scores, they were considered as highly susceptible. The infection type classification was included in the Materials and Methods section.*

L197: 477 or 478?

Reply: *The number represents the position of the mutated amino acids. Revised in the manuscript.*

L222 representative of what?

Reply: *Unclear expression for wheat and Aegilops accessions used for the whole WTK3 gene sequence analysis. Revised in the manuscript.*

Please provide the names of the 20 accessions used for WTK re-sequencing.

Reply: *Detail information of the 25 accessions for entire WTK gene re-sequencing was updated in the main text and supplementary Figure 5. Revised also in the manuscript.*

L240 Authors said that accession Hongmangmai carries as well the 6-bp deletion and it is highly resistant to powdery mildew. No phenotypic evaluation of this accession is included in the manuscript. I would appreciate to see data against a set of isolates (if this is not already done) as well as the entire Pm24 CDS sequence of this accession (to make sure it share the same haplotype with the three other resistant).

Reply: *Due to time limitation, we only tested the reaction of Hongmangmai to 36 Bgt isolates. The infection type of Hongmangmai to these Bgt isolates was updated in Supplementary Table 1 and Supplementary Figure 1. The entire genomic DNA sequence of the WTK3 in Hongmangmai was also provided in supplementary Figure 5. The CDS of WTK3 in Hongmangmai is also identical to that of HLT, BHL and CYC (Supplementary Figure 5).*

The Mat and Meth section is very poor. There is a lot of missing information. In addition, order of methods in the Mat and Meth section is different from how they occur in the main text. So please correct this.

Reply: *More details were provided in the revised Materials and Methods section to provide more information related to the performed research.*

Missing information:

L379 "with some modifications" Which ones? Please indicate how many plants have been scored per isolate.

Reply: *Revised as suggested. We have at least 20 individual plants for a family (fine mapping, mutagenesis and transgenic assays) tested for their reaction to Bgt isolate E09. The reactions of HLT, CYC and BHL to 93 Bgt isolates were scored for 3 plants per isolate. Information was updated in the revised manuscript.*

L410 PCR condition for WTK gene re-sequencing.

Reply: *Revised in the Materials and Method section.*

L415 please indicate a reference

Reply: Revised as suggested.

How genotyping assays were performed?

Reply: Revised in the Materials and Method section.

Which Chinese spring genome sequence

Reply: We used Chinese Spring assembly Ver. 1.0. Revised as suggested.

L419-420: a set of different tools were used for annotation but how annotations from different tools were merged?

Reply: Revised as suggested. "Genes were annotated with the TriAnnot pipeline (<https://urgi.versailles.inra.fr/triannot/?pipeline>) and confirmed using the BLAST analysis tools available at NCBI (<https://www.ncbi.nlm.nih.gov>) and EnsemblPlants (<http://plants.ensembl.org>)."

L427/428 please indicate the primers used.

Reply: Revised as suggested.

L431 PCR amplification, please detail.

Reply: Revised in the Materials and Methods section.

There is no explanation of how genes candidate expression was performed. How they have been cloned and sequenced?

Reply: Revised in the Materials and Methods section.

How many plants per T1 family have been evaluated?

Reply: Revised as suggested. Detail information for all the transgenic lines was provided in Supplementary Table 3.

Nothing about re-sequencing (PCR condition) and haplotype comparison.

Reply: Revised in the Materials and Methods section.

Could authors indicate the specific condition used to amplify the STS marker?

Reply: Revised in the Materials and Methods section.

Figure 1:

The title is Map-based cloning of Pm24. In the main text it is referred as Map-based cloning of MIHLT. At this step authors are not 100% sure that both genes are allelic I would keep MIHLT in Figure 1 as well. Remove Pm24 from the legend and change by MIHLT.

Reply: Revised as suggested.

N=3720, it can be confusing between number of gametes and real number of individuals, I would replace it by 3,720 F2

Reply: *Revised as suggested.*

The scales used for c, e and f are not exact according to the distance indicated in the main text.

Reply: *Revised as suggested.*

There is a gap between scaffold 3178.1 and scaffold3268.1 that you do not indicate in e. What does this mean?

Reply: *We checked the physical map of Aegilops tauschii AL8/78 and found that the scaffold 3178.1 and scaffold3268.1 are overlapped in the sequence assembly. There is no gap between the two BAC contigs in the final sequence assembly. Figure 1 was updated to correct this problem.*

In the legend, please indicate the mapping population used for genetic mapping, the origin of the physical map and which Chinese spring genome sequence was used. Please explain what are RLK, WTK, HP...

Reply: *Revised as suggested.*

Figure 2

Same remark as Figure 1 for the title same for SM Figure2 Figure4

I would add “genomic or native” sequence of the WTK gene.

Above the pictures is written CNL-CPL1. and in the text CPL-CNL It is confusing. Please correct.

Reply: *Revised as suggested.*

Figure 3

Could authors indicate which accessions belong to the different categories (common wheat landraces, mini core collection...)

Reply: *Revised as suggested. Detail information was updated in Supplementary Table 5.*

How was done the clustering of haplotypes?

Reply: *The entire genomic DNA sequences of WTK3 gene (10,410 bp) for 24 Aegilops tauschii and common wheat accessions were compared to have a major picture of the sequence variation of the WTK3 gene. Four major groups were classified as listed in Figure 3a and Supplementary Figure 5. We analyzed the sequence variation of the critical 6-bp deletion region (632 bp) in 1,069 diversified Aegilops tauschii and common wheat accessions and found 10 haplotypes for this region (Figure 3b, Supplementary Figure 6). The information was revised in the manuscript.*

In “a” please indicate where is the 6-bp deletion located.

Reply: The position of the 6-bp deletion was indicated in Figure 3a, 3b and Supplementary Figure 5 and 6.

Please clarify the legends about the position of SNP, deletion and insertion. Which accession is the reference?

Reply: Revised as suggested. The sequence of WTK3 from HLT was used as reference.

I'm surprised to see that HLT and Shi4185 belong to the same haplotypic group. Can you explain?

Reply: Since the entire WTK3 genomic sequence is quite long (10,410 bp), we only re-sequenced 24 accessions of *Aegilops tauschii* and common wheat. Based on large sequence variations (InDels), the 24 accessions were classified as 4 major groups. HLT and Shi4185 belong to the same major group, but with sequence variation (one SNP and one InDel, Fig. 1, Supplementary Figure 5). Since the 6-bp deletion is critical to the powdery mildew resistance, we then tested a 632 bp genomic region of WTK3 covering the 6-bp deletion in 1,069 world wide diversified *Aegilops tauschii* and common wheat collections. Ten haplotypes were found in this genomic region (Fig. 3b, Supplementary Figure 6). The related content was revised as suggested.

What represent the nucleotides in red? Please indicate in the legend.

Reply: Revised as suggested.

Figure 4

There is no LB and RB in the figure compared to the legend.

Reply: Revised as suggested.

SM Figure 2

There is no information about the number of replicates. Please perform statistical analysis to see if the differences in expression are significant along the time course.

Reply: Revised as suggested.

SM3

Please indicates what the numbers in brackets are.

There is a typo in the legend.

Reply: Revised as suggested.

Please indicate clearly in the legend the differences between the variants, e.g. fourth intron not spliced....

Reply: Revised as suggested.

SM figure 4

There is a shift between the names of the sequences and the sequences which makes the figure difficult to read.

Reply: *Revised as suggested.*

I'm not quite sure to understand why you said that "Pm24 protein structure was similar with Yr15". Here first you look at the protein sequence and it is therefore difficult to talk about structure and I observed many differences between both sequences.

Reply: *Sorry for the misinterpretation of the result. Revised in the manuscript.*

Please explain "KinI"

Reply: *Kin I represents Kinase domain I of the Tandem kinase protein (TKP). Revised in the manuscript.*

I think it would have been better to separate domain I and domain II in two different alignments.

Reply: *The Kin I and Kin II were aligned together to compare the conserved residues of the protein kinase domains in Supplementary Figure 4. The divergence residues in the conserved key residues of subdomains of Kin I and Kin II could help us to predict the putative kinase or pseudokinase of a kinase domain.*

Mut614 G718R is located on a position where there is no G. Please correct. Same for Mut477

Reply: *The Mut614 has a G718R missense mutation and located on a position where there is a D for most of the kinase domains. However, several kinases like BAK1 and Stpk-v have a G at this position, same as WTK3. The G718R mutation may also result susceptibility of the WTK3 gene.*

Mut477 P835S is located on a position where there is a conserved P (subdomain XI). The P835S mutation resulted susceptibility of the WTK3 gene, indicating the Kin II domain of WTK3 is also important for the WTK 3 function.

SM Table 4

Please provide information about the isolate used to obtain the IF. What does CWMCC mean?

Reply: *The information on the isolates used to obtain the infection type of HLT, BHL, HLT and Hongmangmai was revised in Supplementary Table 1 and Supplementary Figure 1. CWMCC was Chinese wheat mini-core collection and was revised as MCC in Supplementary Table 5. Information was provided in the revised Supplementary Table 5.*

The title mentioned distribution of haplotypes but there is no information about haplotypes.

Reply: *Revised in Supplementary Table 5 as suggested.*

REVIEWERS' COMMENTS:

Reviewer #1 (Remarks to the Author):

First of all, I would like to apologize to the authors for taking an exceptionally long time to complete my review of the revised manuscript!

I have now been through the manuscript again and looked at the changes. My concerns have largely been addressed. I concur with the authors regarding removing the preliminary work on protein localisation.

The statements regarding broad-spectrum resistance (which was also a concern of the other reviewer) have been toned down or removed. However, the authors might have considered adding a qualifier following lines 354 to 356: With the present assay, the authors cannot formally rule out that there are Bgt isolates which actually overcome Pm24/WTK3, but which are held back by other resistance genes in the Pm24 line tested.

Another concern I raised has to do with the many constructs which were generated as part of this study. For future manuscripts I would strongly suggest that constructs be deposited in the free, public repository Addgene. This ensures that these important resources and outputs are curated and made democratically available to all scientists across the world. In turn, this increases the reproducibility and impact of our work.

Apart from this, I noticed a few minor things:

1. I would suggest tweaking the title to:

A rare gain of function mutation in a wheat tandem kinase confers resistance to powdery mildew

This, I think, is more grammatically correct than the current title.

Lines 350 to 351: I would suggest tweaking the sentence to read:

"Yr15, Rpg1, and Un8 have been shown to provide a broad-spectrum of disease resistance to their respective biotrophic fungal pathogens"

Lines 351 to 354: I would suggest tweaking the grammar so that the sentence reads:

"However, the race specific stem rust resistance gene Sr60 conferred intermediate level of resistance to 3 of 8 tested Pgt races by delaying but not stopping the Pgt infection; therefore it is classified as a partial resistance gene"

Line 205: This should read: "...suggesting possible mutations in other unknown..."

Reviewer #2 (Remarks to the Author):

Authors have taken into account some of the previous reviews. However, there are still a significant numbers of reviews without modifications (while it is mentioned "revised" in the reply).

- The authors can't say in the discussion that "Pm24 that confers resistance to all of the 93 bgt isolates (L300)". This is not true as there might be other genes that control resistance to these isolates in the wheat carrying Pm24. If they would like to tell that Pm24 confers resistance against these isolates it would require evaluating their transgenics (positive and negative sister lines) against these 93 isolates and not only against one isolate E09. I suggest testing their transgenic with a certain number of isolates to provide evidences that it confers broad-spectrum resistance.

Reply: Many thanks for the suggestions. In the revised manuscript, we rephrased the statement of "broad-spectrum". The 93 Bgt isolates were re-sequenced genetically divergent Bgt collections in China and used for genome-wide association study of AvrPm3 (McNally et al. 2018 New Phytol. 218:681-695). A citation of this publication (Ref. 27) was provided in the revised manuscript. Since we need more phenotypic data for the Pm24 gene to more diversified Bgt isolates inoculation, we just mentioned "In the current study, we showed that Pm24 confers resistance to all of the available 93 divergent Bgt isolates collected from 12 provinces of China, suggesting that WTK3 may also be a putative broad-spectrum resistance protein."

Reply: I still do not agree with this last statement. Are you sure that Chiyacao, Baihulu and Hulutou carries only Pm24. They may carry other Pm genes that would explain the resistance against the 93 isolates. Without testing the 93 isolates against your transgenics (positive and negative) or testing the 93 isolates against your F2 populations, it is not right to say that Pm24 confers resistance to 93 isolates. You may say that the Pm24 carrying accessions Baihulu,.. are resistant to 93 isolates.

Minor comments:

- Are WGG markers SNP or SSR? please provide this information in the document.
- Names of Bgt isolates are not nearly identical between SMF1 and SMT1, Please correct
- L194 I don't understand why SMF4 is referenced here.
- L200 Please change Pm24 to MIHLT.
- L227 and in the same paragraph: WTK3 instead of WTK?
- SMT5 632 bp instead of 672?
- L372 it is not orthologs but homeologs.
- L377 it is not homeologs but orthologous
- L379 barley chromosomes are not homeologous of wheat chromosomes
- SMF2 please used MILHT instead of Pm24, Please indicate the statistical test used to evaluate differences between samples.
- Fig1a is not correctly referenced and not even described in the main document.
- SMF5 and SMF6: This is not easy to identify mutations in these alignments. Could you please provide alignments were only differences are shown?
- L247 marker InDel-WTK3 (the name in the main text and SMT2 are different.

There is still missing information in the Materiel and Methods and section.

L379 "with some modifications" Which ones? Please indicate how many plants have been scored per isolate.

Reply: Revised as suggested. We have at least 20 individual plants for a family (fine mapping, mutagenesis and transgenic assays) tested for their reaction to Bgt isolate E09. The reactions of HLT, CYC and BHL to 93 Bgt isolates were scored for 3 plants per isolate. Information was updated in the revised manuscript

Reply: I can't see the information of 3 plants per isolate in the Mat and Meth section.

How genotyping assays were performed?

Reply: Revised in the Materials and Method section.

Reply: I can't find this information in the Materiel and Method section.

L431 PCR amplification, please detail.

Reply: Revised in the Materials and Methods section

Reply: there is still no information provided regarding the PCR amplification (Master mix or Taq,

annealing temperature). Neither for mutant analysis.

Figure2 Same remark as Figure 1 for the title same for SM Figure2 Figure4 I would add "genomic or native" sequence of the WTK gene. Above the pictures is written CNL-CPL1. and in the text CPL-CNL It is confusing. Please correct.

Reply: Revised as suggested.

Reply: there is no modification in the title.

Could authors indicate the score on Figure 2 and SM table 3 for the mutants as it seems that the level of susceptibility is different?

Reply: The infection type (IT) of the mutants were classified as 4 (highly susceptible) according to a 0, 0;, 1 to 4 scales, representing immune, necrosis flecks, highly resistant, moderate resistant, moderate susceptible, and highly susceptible. Minor different between 3 and 4 were observed for some EMS mutants. However, based on the M3 families IT scores, they were considered as highly susceptible. The infection type classification was included in the Materials and Methods section Please provide this information in the legends of Figure 2.

L197: 477 or 478?

Reply: The number represents the position of the mutated amino acids. Revised in the manuscript.

Reply: the number is different between SMT4 and the main text.

Dear Reviewers,

Thank you very much for your time in reviewing our manuscript and we appreciate the great comments in helping us to improve the revised manuscript. Enclosed please find our responses for your comments and we also revised the relative section according to your nice suggestions.

Thanks again

Zhiyong Liu

REVIEWERS' COMMENTS:

Reviewer #1 (Remarks to the Author):

First of all, I would like to apologize to the authors for taking an exceptionally long time to complete my review of the revised manuscript!

I have now been through the manuscript again and looked at the changes. My concerns have largely been addressed. I concur with the authors regarding removing the preliminary work on protein localisation.

The statements regarding broad-spectrum resistance (which was also a concern of the other reviewer) have been toned down or removed. However, the authors might have considered adding a qualifier following lines 354 to 356: With the present assay, the authors cannot formally rule out that there are *Bgt* isolates which actually overcome *Pm24/WTK3*, but which are held back by other resistance genes in the *Pm24* line tested.

Response: Many thanks for the suggestion. We revised the sentence as “In the current study, we showed that the *Pm24* carrying accessions CYC, BHL, HLT and HMM are resistant to all of the tested 93 *Bgt* isolates collected from 12 provinces of China. With the present assay, we cannot formally rule out that there are *Bgt* isolates which actually overcome *Pm24/WTK3*, but which are held back by other resistance genes in the lines carrying *Pm24/WTK3*.”.

Another concern I raised has to do with the many constructs which were generated as part of this study. For future manuscripts I would strongly suggest that constructs be deposited in the free, public repository Addgene. This ensures that these important resources and outputs are curated and made democratically available to all scientists across the world. In turn, this increases the reproducibility and impact of our work.

Response: Thanks for the suggestion. We can't create an account at the Addgene website these days due to unknown reason. We contacted with the Addgene (email attached) and will deposit the constructs into the public repository Addgene when it works again. Also, the constructs are also freely available upon request.

Apart from this, I noticed a few minor things:

1. I would suggest tweaking the title to:

A rare gain of function mutation in a wheat tandem kinase confers resistance to powdery mildew

This, I think, is more grammatically correct than the current title.

Response: Revised as suggested.

Lines 350 to 351: I would suggest tweaking the sentence to read:
"Yr15, Rpg1, and Un8 have been shown to provide a broad-spectrum of disease resistance to their respective biotrophic fungal pathogens"

Response: Revised as suggested.

Lines 351 to 354: I would suggest tweaking the grammar so that the sentence reads:

"However, the race specific stem rust resistance gene Sr60 conferred intermediate level of resistance to 3 of 8 tested Pgt races by delaying but not stopping the Pgt infection; therefore it is classified as a partial resistance gene"

Response: Revised as suggested.

Line 205: This should read: "...suggesting possible mutations in other unknown..."

Response: Revised as suggested.

Reviewer #2 (Remarks to the Author):

Authors have taken into account some of the previous reviews. However, there are still a significant numbers of reviews without modifications (while it is mentioned "revised" in the reply).

- The authors can't say in the discussion that "Pm24 that confers resistance to all of the 93 bgt isolates (L300)". This is not true as there might be other genes that control resistance to these isolates in the wheat carrying Pm24. If they would like to tell that Pm24 confers resistance against these isolates it would require evaluating their transgenics (positive and negative sister lines) against these 93 isolates and not only against one isolate E09. I suggest testing their transgenic with a certain number of isolates to provide evidences that it confers broad-spectrum resistance.

Reply: Many thanks for the suggestions. In the revised manuscript, we rephrased the statement of "broad-spectrum". The 93 Bgt isolates were re-sequenced genetically divergent Bgt collections in China and used for genome-wide association study of AvrPm3 (McNally et al. 2018 New Phytol. 218:681-695). A citation of this publication (Ref. 27) was provided in the revised manuscript. Since we need more phenotypic data for the Pm24 gene to more diversified Bgt isolates inoculation, we just mentioned "In the current study, we showed that Pm24 confers resistance to all of the available 93 divergent Bgt isolates collected from 12 provinces of China, suggesting that WTK3 may also be a putative broad-spectrum resistance protein."

Reply: I still do not agree with this last statement. Are you sure that Chiyacao, Baihulu and Hulutou carries only Pm24. They may carry other Pm genes that would explain the resistance against the 93 isolates. Without testing the 93 isolates against your transgenics (positive and negative) or testing the 93 isolates against your F2 populations, it is not right to say that Pm24 confers resistance to 93 isolates. You may say that the Pm24 carrying accessions Baihulu,.. are resistant to 93 isolates.

Response: Many thanks for the critical suggestion. We cannot rule out that there are *Bgt* isolates which actually overcome *Pm24/WTK3* in the 4 Chinese landrace CYC, BHL, HLT and HMM. We revised the sentence as "In the current study, we showed that the *Pm24* carrying accessions CYC, BHL, HLT and HMM are resistant to all of the tested 93 *Bgt* isolates collected from 12 provinces of China. With the present assay, we cannot formally rule out that there are *Bgt* isolates which actually overcome *Pm24/WTK3*, but which are held back by other resistance gens in the lines carrying *Pm24/WTK3*."

Minor comments:

- Are WGG markers SNP or SSR? please provide this information in the document.

Response: The marker type details of the WGG markers were provided in the revised Supplementary Table 2.

- Names of Bgt isolates are not nearly identical between SMF1 and SMT1, Please correct

Response: We have only 36 *Bgt* isolated tested for the four Pm24 carrying wheat landrace (CYC, BHL, HLT and HMM) at the same time (Supplementary Figure 1). The reactions (Infection types) of CYC, BHL, and HLT to other 57 *Bgt* isolates were recorded at the seedling stage. Reaction of HMM to the 57 *Bgt* isolates was not tested after it was identified from the wheat landrace genebank. The names of the 36 *Bgt* isolates in SMT1 and SMF1 were corrected to keep them identical.

- L194 I don't understand why SMF4 is referenced here.

Response: SMF4 is removed from the revised manuscript.

- L200 Please change Pm24 to MIHLT.

Response: Revised as suggested.

- L227 and in the same paragraph: WTK3 instead of WTK?

Response: Revised as suggested.

- SMT5 632 bp instead of 672?

Response: Revised as suggested.

- L372 it is not orthologs but homeologs.

Response: Revised as "Homeologs of *WTK3* ..." as suggested.

- L377 it is not homeologs but orthologous

Response: Revised as "suggesting that the orthologous copy of *WTK3*..." as suggested.

- L379 barley chromosomes are not homeologous of wheat chromosomes

Response: Revised as "orthologous chromosome of *Hordeum vulgare*" as suggested.

- SMF2 please use MIHLT instead of Pm24, Please indicate the statistical test used to evaluate differences between samples.

Response: Revised as suggested. Student's t test was used to evaluate the difference between samples.

- Fig1a is not correctly referenced and not even described in the main document.

Response: Revised as "The *MIHLT* locus was previously mapped to a 3.6 cM interval flanked by markers *Xwggc3026* and *Xwggc3148* in the terminal region of chromosome arm 1DS²⁶ using a mapping population developed between the powdery mildew resistant HLT and highly susceptible cultivar Shi 4185 (S4185) (Fig. 1a, 1b). S4185 was highly susceptible to *Bgt* isolate E09 with a large number of visible conidia produced at 10 dpi and HLT was highly resistant to *Bgt* isolate E09 with no visible conidia produced (Fig. 1a). Trypan blue and DAB staining also showed large number of spores produced in S4185 and robust accumulation of H₂O₂ in HLT (Fig. 1a)."

- SMF5 and SMF6: This is not easy to identify mutations in these alignments. Could you please provide alignments where only differences are shown?

Response: The genomic DNA sequences and CDS of the *WTK3* are identical in Hulutou (HLT), Baihulu (BHL), Chiyacao (CYC) and Hongmangmai (HMM), respectively. Therefore, there is no SNP or InDel for the 4 landraces presented in SMF5. The SNPs and InDels in the sequence alignments of 25 wheat and *Aegilops tauschii* accessions were shown in different colors at revised SMF6.

- L247 marker InDel-WTK3 (the name in the main text and SMT2 are different).

Response: Revised as "InDel-WTK3" in main text and SMT2.

There is still missing information in the Materials and Methods and section.

L379 "with some modifications" Which ones? Please indicate how many plants have been scored per isolate.

Reply: Revised as suggested. We have at least 20 individual plants for a family (fine mapping, mutagenesis and transgenic assays) tested for their reaction to *Bgt* isolate E09. The reactions of HLT, CYC and BHL to 93 *Bgt* isolates were scored for 3 plants per isolate. Information was updated in the revised manuscript

Reply: I can't see the information of 3 plants per isolate in the Materials and Methods section.

Response: The information was revised as "Three individual plants of CYC, BHL, HLT, HMM and Chancellor were inoculated with 36 *Bgt* isolates (Supplementary Fig. 1). At least 20 individuals of the M₂ and T₁ transgenic families were inoculated with *Bgt* isolate E09. For fine mapping of *MIHLT*, 25-30 plants of the F₃ families were screened for reaction to *Bgt* isolate E09 to confirm the genotypes of the corresponding F₂ plants." in the method section.

How genotyping assays were performed?

Reply: Revised in the Materials and Methods section.

Reply: I can't find this information in the Materials and Methods section.

Response: The genotyping of the fine mapping population and recombinants was revised in the Method. Details of the PCR and gel electrophoresis were provided.

L431 PCR amplification, please detail.

Reply: Revised in the Materials and Methods section

Reply: there is still no information provided regarding the PCR amplification (Master mix or Taq, annealing temperature). Neither for mutant analysis.

Response: The genotyping of the fine mapping population and recombinants was revised in the Method. Details of the PCR and gel electrophoresis were provided in the revised Method section.

Figure2 Same remark as Figure 1 for the title same for SM Figure2 Figure4 I would add "genomic or native" sequence of the WTK gene. Above the pictures is written CNL-CPL1. and in the text CPL-CNL It is confusing. Please correct.

Reply: Revised as suggested.

Reply: there is no modification in the title.

Response: In Figure 1g and h, the description was revised as "g and h Difference in the structures and genomic sequences of *WTK3* and *CNL* genes between HLT and S4185, respectively."

Could authors indicate the score on Figure 2 and SM table 3 for the mutants as it seems that the level of susceptibility is different?

Reply: The infection type (IT) of the mutants were classified as 4 (highly susceptible) according to a 0, 0;, 1 to 4 scales, representing immune, necrosis flecks, highly resistant, moderate resistant, moderate susceptible, and highly susceptible. Minor different between 3 and 4 were observed for some EMS mutants. However, based on the M3 families IT scores, they were considered as highly susceptible. The infection type classification was included in the Materials and Methods section

Please provide this information in the legends of Figure 2.

Response: Revised in the legends of Figure 2, 4 and Supplementary Table 4.

L197: 477 or 478?

Reply: The number represents the position of the mutated amino acids. Revised in the manuscript.

Reply: the number is different between SMT4 and the main text.

Response: Revised in the SMT4. The right position of this pre-mature stop codon is at the 478 position.